# ATTENTION-GUIDED CONTRASTIVE ROLE REPRESENTATIONS FOR MULTI-AGENT REINFORCEMENT LEARNING

**Zican Hu**[1], **Zongzhang Zhang**[2], **Huaxiong Li**[1], **Chunlin Chen**[1], **Hongyu Ding**[1], **Zhi Wang**[1][*]

[1] Department of Control Science and Intelligent Engineering, Nanjing University
[2] School of Artificial Intelligence, Nanjing University
`{zicanhu,hongyuding}@smail.nju.edu.cn`
`{zzzhang,huaxiongli,clchen,zhiwang}@nju.edu.cn`

## ABSTRACT

Real-world multi-agent tasks usually involve dynamic team composition with the emergence of roles, which should also be a key to efficient cooperation in multi-agent reinforcement learning (MARL). Drawing inspiration from the correlation between roles and agent's behavior patterns, we propose a novel framework of **A**ttention-guided **CO**ntrastive **R**ole representation learning for **M**ARL (**ACORM**) to promote behavior heterogeneity, knowledge transfer, and skillful coordination across agents. First, we introduce mutual information maximization to formalize role representation learning, derive a contrastive learning objective, and concisely approximate the distribution of negative pairs. Second, we leverage an attention mechanism to prompt the global state to attend to learned role representations in value decomposition, implicitly guiding agent coordination in a skillful role space to yield more expressive credit assignment. Experiments on challenging StarCraft II micromanagement and Google research football tasks demonstrate the state-of-the-art performance of our method and its advantages over existing approaches. Our code is available at https://github.com/NJU-RL/ACORM.

## 1 INTRODUCTION

Cooperative multi-agent reinforcement learning (MARL) aims to coordinate a system of agents towards optimizing global returns (Vinyals et al., 2019), and has witnessed significant prospects in various domains, such as autonomous vehicles (Zhou et al., 2020), smart grid (Chen et al., 2021a), robotics (Yu et al., 2023), and social science (Leibo et al., 2017). Training reliable control policies for coordinating such systems remains a major challenge. The centralized training with decentralized execution (CTDE) (Foerster et al., 2016) hybrids the merits of independent Q-learning (Foerster et al., 2017) and joint action learning (Sukhbaatar et al., 2016), and becomes a compelling paradigm that exploits the centralized training opportunity for training fully decentralized policies (Wang et al., 2023). Subsequently, numerous popular algorithms are proposed, including VDN (Sunehag et al., 2018), QMIX (Rashid et al., 2020), MAAC (Iqbal & Sha, 2019), and MAPPO (Yu et al., 2022).

Sharing policy parameters is crucial for scaling these algorithms to massive agents with accelerated cooperation learning (Fu et al., 2022). However, it is widely observed that agents tend to acquire homogeneous behaviors, which might hinder diversified exploration and sophisticated coordination (Christianos et al., 2021). Some methods (Li et al., 2021; Jiang & Lu, 2021; Liu et al., 2023) attempt to promote individualized behaviors by distinguishing each agent from the others, while they often neglect the prospect of effective team composition with implicit task allocation. Real-world multi-agent tasks usually involve dynamic team composition with the emergence of roles (Shao et al., 2022; Hu et al., 2022). [1] Early works introduce the role concept into multi-agent

---

[*]Correspondence to Zhi Wang <zhiwang@nju.edu.cn>.

[1]Taking the football game (Kurach et al., 2020) as an example, the midfielders are primarily responsible for delivering the ball to the forwards to coordinate shots on goal in the offensive phase, while they need to drop back and join the defenders to block passing lanes on the defensive.

systems (Dastani et al., 2003; Sims et al., 2008; Lhaksmana et al., 2018), while they usually require prior domain knowledge to pre-define role responsibilities. Recently, ROMA (Wang et al., 2020) learns emergent roles conditioned solely on current observations, and RODE (Wang et al., 2021) associates each role with a fixed subset of the joint action space. COPA (Liu et al., 2021) allows dynamic role allocation via distributing a global view of team composition to each agent during execution. Some works decompose the task into a set of skills (Liu et al., 2022) or subtasks (Yang et al., 2022; Iqbal et al., 2022) with a hierarchical structure for control. Overall, existing role-based methods still suffer from several deficiencies, such as insufficient characterization of complex behaviors for role emergence, neglect of evolving team dynamics, or relaxation of the CTDE constraint.

To better leverage dynamic role assignment, we propose a novel framework of **A**ttention-guided **CO**ntrastive **R**ole representation learning for **M**ARL (**ACORM**). Our main insight is to learn a compact role representation that can capture complex behavior patterns of agents, and use that role representation to promote behavior heterogeneity, knowledge transfer, and skillful coordination across agents. First, we formalize the learning objective as mutual information maximization between the role and its representation, to maximally reduce role uncertainty given agent's behaviors while minimally preserving role-irrelevant information. We introduce a contrastive learning method to optimize the infoNCE loss, a mutual information lower bound. To concisely approximate the distribution of negative pairs, we extract agent behaviors by encoding its trajectory into a latent space, and periodically partition all agents into several clusters according to their latent embeddings where points from different clusters are paired as negative. Second, during centralized training, we employ an attention mechanism to prompt the global state to attend to learned role representations in value decomposition. The attention mechanism implicitly guides agent coordination in a skillful role space, thus yielding more expressive credit assignment with the emergence of roles. [2]

ACORM is fully compatible with CTDE methods, and we realize ACORM on top of two popular MARL algorithms, QMIX (Rashid et al., 2020) and MAPPO (Yu et al., 2022), benchmarked on challenging StarCraft multi-agent challenge (SMAC) (Samvelyan et al., 2019) and Google research football (GRF) (Kurach et al., 2020) environments. Experiments demonstrate that ACORM achieves state-of-the-art performance on most scenarios. Visualizations of learned role representations, heterogeneous behavior patterns, and attentional value decomposition shed further light on our advantages. Ablation studies confirm that ACORM promotes higher coordination capacity by virtue of contrastive role representation learning and attention-guided credit assignment, respectively, even if agents have the same innate characteristics. In summary, our contributions are threefold:

- We propose a general role representation learning framework based on contrastive learning, which effectively tackles agent homogenization and facilitates efficient knowledge transfer.
- We leverage role representations to realize more expressive credit assignment via an attention mechanism, promoting strategical coordination in a sophisticated role space.
- We build our method on top of popular QMIX and MAPPO, and conduct extensive experiments on SMAC and GRF to demonstrate our state-of-the-art performance and advantages.

## 2 METHOD

In this section, we present the ACORM framework. We consider cooperative multi-agent tasks formulated as a Dec-POMDP (Oliehoek & Amato, 2016), $G = \langle I, S, A, P, R, \Omega, O, n, \gamma \rangle$, where $I$ is a finite set of $n$ agents, $s \in S$ is the global state, and $\gamma \in [0, 1)$ is the discount factor. At each time step, each agent $i$ draws an observation $o_i \in O$ from $\Omega(s, i)$ and selects a local action $a_i \in A$. After executing the joint action $\boldsymbol{a} = [a_1, ..., a_n]^\top \in A^n$, the system transitions to a next state $s'$ according to $P(s'|s, \boldsymbol{a})$ and receives a reward $r = R(s, \boldsymbol{a})$ shared by all agents.

Our idea is to learn a compact role representation that can characterize complex behavior patterns of agents, and use the role information to facilitate individual policy learning and guide agent coordination. Agents with similar roles can enjoy higher learning efficiency via more aggressive knowledge transfer, and agent heterogeneity is also guaranteed with the discrimination of diverse roles. Formally, we propose the following definition of the role and its representation.

---

[2]Taking StarCraft II as an example, the acquired roles represent diverse strategies in a team-based manner, such as focusing fire, sneaking attack, and drawing fire. Agents with similar role representations learn a specialized strategy more efficiently by more positive information sharing, and the attention mechanism is responsible for coordinating these heterogeneous behaviors more strategically with clearer role extraction in the team.

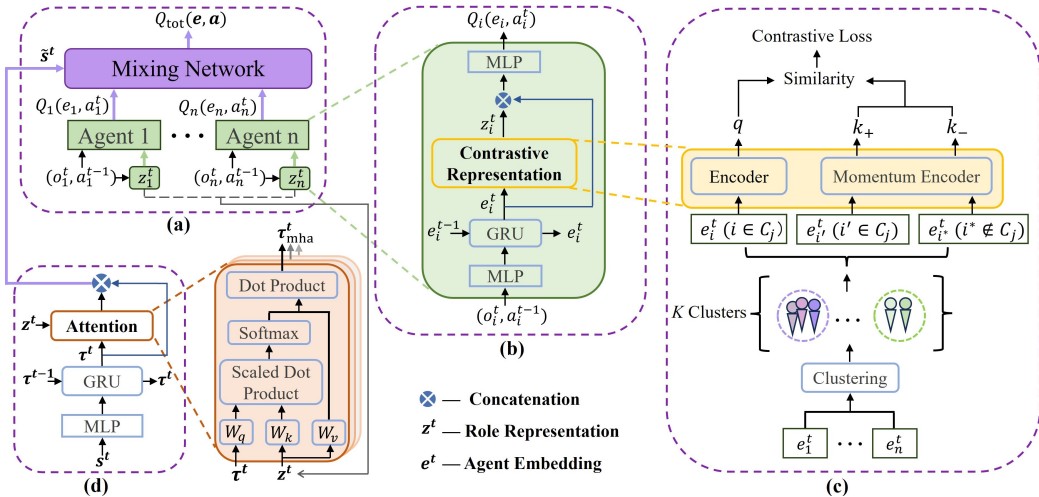

Figure 1: The ACORM framework based on QMIX. (a) The overall architecture. (b) The structure of shared individual Q-network. (c) The detail of contrastive role representation learning, where $z_i$ is the query $q$, and $z_{i'}/z_{i*}$ are positive/negative keys $k_+/k_-$. (d) The attention module that incorporates learned role representations into the mixing network's input for better value decomposition.

**Definition 1.** *Given a cooperative multi-agent task $G = \langle I, S, A, P, R, \Omega, O, n, \gamma \rangle$, each agent $i$ is associated with a role $M_i \in \mathcal{M}$ that describes its behavior pattern. Each role $M_i$ is quantified by a role representation $z_i \in \mathcal{Z}$, which is obtained by training a complex function as $z_i = f(\rho_i)$, where $\rho_i \in \Gamma \equiv (O, A)^l$ is the local trajectory of agent $i$, and $l$ is the number of observation-action pairs. $\pi_{z_i} : O \times A \times \mathcal{Z} \rightarrow [0, 1]$ is the individual policy for agent $i$.*

ACORM consists of individual Q-networks in Fig. 1(b) and a mixing network in Fig. 1(a). We introduce mutual information maximization to formalize role representation learning, and derive a contrastive learning objective that optimizes agent embeddings $\{e_i^t\}_{i=1}^n$ in a self-supervised way to acquire contrastive role representations $\{z_i^t\}_{i=1}^n$, which is shown in Fig. 1(c) and will be introduced in detail in Section 2.1. In value decomposition, we employ multi-head attention (MHA) to prompt the global state to attend to learned role patterns, guiding skillful agent coordination in the high-level role space for facilitating expressive credit assignment, as described in Fig. 1(d) and in Section 2.2. Appendix B presents the pseudocode, and Appendix C gives the extension to MAPPO.

## 2.1 CONTRASTIVE ROLE REPRESENTATIONS

Our objective is to ensure that agents with similar behavior patterns exhibit closer role representations, while those with notably different strategies are pushed away from each other. This stands in contrast to using a one-hot ID to preserve the agent's individuality, which lacks adequate discrimination under the paradigm of parameter sharing. Hence, the primary issues we aim to tackle are: i) how to define a feasible metric to quantify the degree of similarity between agent's behaviors, and ii) how to develop an efficient method to optimize the discrimination of role representations.

**Agent Embedding.** To tackle the first issue, we learn an agent embedding $e_i^t$ from each agent's trajectory to extract complex agent behaviors with contextual knowledge as $e_i^t = f_\phi(o_i^t, a_i^{t-1}, e_i^{t-1})$, where $\phi$ is a shared gated recurrent unit (GRU) encoder, $o_i^t$ is the current observation, $a_i^{t-1}$ is the last action, and $e_i^{t-1}$ is the hidden state of the GRU. Naturally, the distance between the obtained agent embeddings can serve as the metric to measure the behavior dissimilarity between agents.

**Contrastive Learning.** An ideally discriminative role representation should be dependent on roles associated with agent's behavior patterns, while remaining invariant across agent identities. We introduce mutual information to measure the mutual dependency between the role and its representation. Formally, mutual information aims to quantify the uncertainty reduction of one random variable when the other one is observed. To tackle the second issue, we propose to maximize the mutual information between the role and its representation, and learn a role encoder that maximally

reduces role uncertainty while minimally preserving role-irrelevant information. Mathematically, we formalize the role encoder $\theta$ as a probabilistic encoder $z^t \sim f_\theta(z^t|e^t)$, where $z^t$ denotes the role representation at time $t$, and $e^t = f_\phi(\sum_{t'=1}^{t}(o^{t'}, a^{t'-1}))$ denotes the agent embedding obtained from the history trajectory. [3] Role $M$ follows the role distribution $P(M)$, and the distribution of agent embedding $e$ is determined by its role. The learning objective for the role encoder is:

$$\max\ I(z; M) = \mathbb{E}_{z,M}\left[\log \frac{p(M|z)}{p(M)}\right]. \tag{1}$$

In practice, directly optimizing mutual information is intractable. Inspired by the noise contrastive estimation (InfoNCE) (Oord et al., 2018) in the literature of contrastive learning (Laskin et al., 2020; Yuan & Lu, 2022), we derive a lower bound of Eq. (1) with the following theorem.

**Theorem 1.** *Let $\mathcal{M}$ denote a set of roles following the role distribution $P(M)$, and $|\mathcal{M}| = K$. $M \in \mathcal{M}$ is a given role. Let $e = f_\phi(\sum_t(o^t, a^{t-1}))$, $z \sim f_\theta(z|e)$, and $h(e, z) = \frac{p(z|e)}{p(z)}$, where $\sum_t(o^t, a^{t-1})$ is the agent's local trajectory following a given policy. For any role $M^* \in \mathcal{M}$, let $e^*$ denote the agent embedding generated by the role $M^*$, then we have*

$$I(z; M) \geq \log K + \mathbb{E}_{\mathcal{M}, z, e}\left[\log \frac{h(e, z)}{\sum_{M^* \in \mathcal{M}} h(e^*, z)}\right]. \tag{2}$$

The proof of this theorem is given in Appendix A. Since we cannot evaluate $p(z)$ or $p(z|e)$ directly, we turn to techniques of NCE and importance sampling based on comparing the target value with randomly sampled negative values. Hence, we approximate $h$ with the exponential of a score function $S(z, z^*)$ that is a similarity metric between latent codes of two examples. We derive a sampling version of the tractable lower bound to be the role encoder's learning objective as

$$\min_\theta \mathcal{L}_K = -\mathbb{E}_{M_i \in \mathcal{M}, (e,e') \sim M_i, z \sim f_\theta(z|e)}\left[\log \frac{\exp(S(z, z'))}{\exp(S(z, z')) + \sum_{M^* \in \mathcal{M} \setminus M_i} \exp(S(z, z^*))}\right], \tag{3}$$

where $\mathcal{M}$ is the set of training roles, $e, e'$ are two instances of agent embeddings sampled from the dataset of role $M_i$, and $z, z'$ are latent representations of $e, e'$. For any role $M^* \in \mathcal{M} \setminus M_i$, $z^*$ is the representation of agent embedding $e^*$ sampled by role $M^*$. Following the literature, we denote $(e, e')_{M_i}$ as a positive pair and denote $\{(e, e^*)\}_{M^* \in \mathcal{M} \setminus M_i}$ as negative pairs. The objective in Eq. (3) optimizes for a $K$-way classification loss to classify the positive pair out of all pairs. Minimizing the InfoNCE loss $\mathcal{L}_K$ maximizes a lower bound on mutual information in Eq. (2), and this bound becomes tighter as $K$ becomes larger. The role encoder ought to extract shared features in agent embeddings of the same role to maximize the score of positive pairs, while capturing essential distinctions across various roles to decrease the score of negative pairs.

**Negative Pairs Generation.** Periodically, we partition all $n$ agents into $K$ clusters $\{C_j\}_{j=1}^{K}$ according to agent embeddings.[4] Naturally, we encourage role representations from the same cluster to stay close to each other, while differing from agents in other clusters. For agent $i$, we denote its role representation $z_i$ as the query $q$, and role representations of other agents as the keys $\mathbb{K} = \{z_1, ..., z_n\} \setminus z_i$. Points from the same cluster as the query, $i \in C_j$, are set as positive keys $\{k_+\}$ and those from different clusters are set as negative $\{k_-\} = \mathbb{K} \setminus \{k_+\}$. In practice, we use bilinear products (Laskin et al., 2020) for the score function in Eq. (3), and similarities between the query and keys are computed as $q^\top W k$, where $W$ is a learnable parameter matrix. The InfoNCE loss in Eq. (3) is rearranged as

$$\mathcal{L}_K = -\log \frac{\exp(q^\top W k_+)}{\exp(q^\top W k_+) + \exp(q^\top W k_-)} = -\log \frac{\sum_{i' \in C_j} \exp(z_i^\top W z_{i'})}{\sum_{i' \in C_j} \exp(z_i^\top W z_{i'}) + \sum_{i^* \notin C_j} \exp(z_i^\top W z_{i^*})}. \tag{4}$$

Following the MoCo method (He et al., 2020), we maintain a query encoder $\theta_q$ and a key encoder $\theta_k$, and use a momentum update to facilitate the key representations' consistency as

$$\theta_k \leftarrow \beta \theta_k + (1 - \beta)\theta_q, \tag{5}$$

where $\beta \in [0, 1)$ is a momentum coefficient, and only parameters $\theta_q$ are updated by backpropagation.

---

[3]Here, we use the superscript $t$ for highlighting the time-evolving property of role representations and relevant variables, and we will partially omit it for simplicity in the below.

[4]In this paper, we simply use K-means (Hartigan & Wong, 1979) based on Euclidean distances between agent embeddings. Moreover, it can be easily extended to more complex clustering methods such as Gaussian mixture models (Bishop, 2006). In Appendix D, we conduct the hyperparameter analysis about the influence of different $K$ values and how to determine the number of clusters automatically.

## 2.2 ATTENTION-GUIDED ROLE COORDINATION

After acquiring agents' contrastive role representations from their *local* information, we introduce an attention mechanism in value decomposition to enhance agent coordination in the sophisticated role space with a *global* view. Popular CTDE algorithms, such as QMIX, realize behavior coordination across agents via a mixing network that estimates joint action-values as a monotonic combination of per-agent values, and the mixing network weights are conditioned on the system's global state. Naturally, it is interesting to incorporate the learned role information into the mixing network to facilitate skillful coordination across roles. The simplest approach is to concatenate the global state and role representations for generating mixing network weights, while it fails to exploit the internal structure to effectively extract correlations in the role space. Fortunately, the attention mechanism (Vaswani et al., 2017) aligns perfectly with our intention by prompting the global state to attend to learned role patterns, thus providing more expressive credit assignment in value decomposition.

The attention mechanism aims to draw global dependencies without regard to their distance in the input or output sequences, and has gained substantial popularity as a fundamental building block of compelling sequence modeling and transduction models, such as GPT (Brown et al., 2020), vision transformers (Dosovitskiy et al., 2021), and decision transformers (Chen et al., 2021b). An attention function can be described as mapping a query and a set of key-value pairs to a weighted sum of the values, where the weight assigned to each value is computed by a compatibility function of the query and corresponding key. As role representations are learned based on extracting agent behaviors from history trajectories, we also use a GRU to encode the history states $(s^0, s^1, ..., s^t)$ into a state embedding $\tau^t$ for facilitating information matching between states and role representations. Then, we set the state embedding $\tau \in \mathbb{R}^{d_s \times d}$ as the query, and the role representations $z = [z_1, ..., z_n]^\top \in \mathbb{R}^{n \times d}$ as both the key and value, where $d$ is the dimension of role representation and $d_s$ is the length of state embedding. Formally, we calculate a weighted combination of role representations as

$$\tau_{\text{atten}} = \sum_{i=1}^n \alpha_i v_i = \sum_{i=1}^n \alpha_i \cdot z_i W^V, \tag{6}$$

where the value $v_i$ is a linear transformation of $z_i$ by a shared parameter matrix $W^V \in \mathbb{R}^{d \times d_v}$. The attention weight $\alpha_i$ computes the relevance between the state embedding $\tau$ and the $i$-th agent's role representation $z_i$, and we apply a softmax function to obtain the weight as

$$\alpha_i = \frac{\exp\left(\frac{1}{\sqrt{d_k}} \cdot \tau W^Q \cdot \left(z_i W^K\right)^\top\right)}{\sum_{j=1}^n \exp\left(\frac{1}{\sqrt{d_k}} \cdot \tau W^Q \cdot \left(z_j W^K\right)^\top\right)}, \tag{7}$$

where $W^Q, W^K \in \mathbb{R}^{d \times d_k}$ are shared parameter matrices for linear transformation of query-key pairs, and $1/\sqrt{d_k}$ is a factor that scales the dot-product attention. We use multi-head attention (MHA) for allowing the model to jointly attend to information from different representation subspaces at different positions, and obtain the aggregated output as

$$\tau_{\text{mha}} = \text{Concat}\left(\tau_{\text{atten}}^1, ..., \tau_{\text{atten}}^H\right) W^O, \tag{8}$$

where $\tau_{\text{atten}}^h$ ($h \in \{1, 2, ..., H\}$) is the attention output using projections of $W_h^Q, W_h^K$, and $W_h^V$, and $W^O \in \mathbb{R}^{H \cdot d_v \times d}$ is the parameter matrix for combining outputs of all heads. Finally, the MHA output is combined with the global state to be responsible for generating weights of the mixing network, as shown in Fig. 1(d). In this way, we flexibly leverage role representations to offer more comprehensive information for value decomposition. By allowing the global state to attend to the learned role patterns, the attention mechanism implicitly guides the agent coordination in a skillful role space, thus yielding more expressive credit assignment with the emergence of roles.

## 3 EXPERIMENTS

We evaluate ACORM to answer the following questions: (i) Can ACORM facilitate learning efficiency and stability in complex multi-agent domains? If so, what are the respective contributions of different modules to the performance gains? (See Sec. 3.1). (ii) Can ACORM learn meaningful role representations associated with agent's behavior patterns and achieve effective dynamic team composition? (See Sec. 3.2). (iii) Can ACORM successfully attend to learned role representations to realize skillful role coordination and more expressive credit assignment? (See Sec. 3.3).

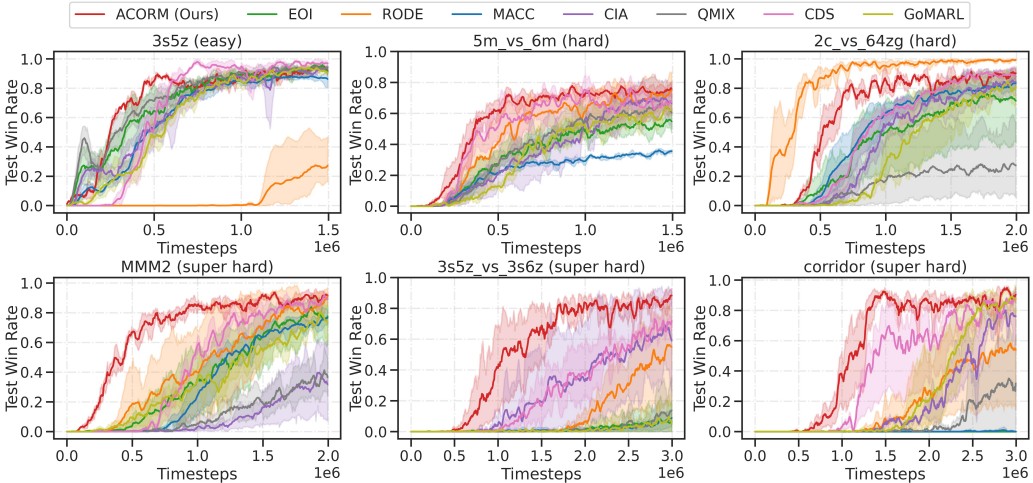

Figure 2: Performance comparison between ACORM and baselines on six representative maps.

**Implementations.** We choose SMAC (Samvelyan et al., 2019) as the first testbed for its rich maps and convenient visualization tools, and realize ACORM on top of the popular QMIX algorithm. For visualization, we render game scenes, and show agent embeddings and role representations using t-SNE. Appendix C gives evaluation results on the GRF benchmark, and Appendix D shows the algorithm architecture, experimental settings and results of MAPPO-based ACORM.

**Baselines.** We compare ACORM to `QMIX` and six baselines: 1) `RODE` (Wang et al., 2021) with action space decomposition; 2) `EOI` (Jiang & Lu, 2021) that encourages diversified individuality via training an observation-to-identity classifier; 3) `MACC` (Yuan et al., 2022) that uses attention to concentrate on most related subtasks; 4) `CDS` (Li et al., 2021) that introduces diversity in both optimization and representation; 5) `CIA` (Liu et al., 2023) that boosts credit-level distinguishability via contrastive learning; and 6) `GoMARL` (Zang et al., 2023) with an automatic grouping mechanism.

### 3.1 PERFORMANCE AND ABLATION STUDY

For evaluation, all experiments are carried out with five different random seeds, and the mean of the test win rate is plotted as the bold line with 95% bootstrapped confidence intervals of the mean (shaded). Appendix B describes the detailed setting of hyperparameters.

**Performance.** SMAC contains three kinds of maps: *easy*, *hard*, and *super hard*. Super hard maps are typically complex tasks that require deeper exploration of diversified behaviors and more skillful coordination. Since ACORM is designed to promote these properties, the performance on these maps is especially significant to validate our research motivation and advantages. Fig. 2 presents the performance of ACORM on six representative tasks, and performance on more maps can be found in Appendix B. ACORM obtains the best performance on all super-hard maps and most of the other maps. A noteworthy point is that ACORM outperforms all baselines by the largest margin on super hard maps that demand a significantly higher degree of behavior diversity and coordination: `MMM2`, `3s5z_vs_3s6z`, and `corridor`. In these maps, ACORM gains an evidently faster-increasing trend regarding the test win rate in the first million training steps, which is supposed to benefit from the high efficiency of exploring cooperatively heterogeneous behaviors via our discriminative role assignment. Moreover, ACORM exhibits the lowest variance in learning curves, signifying not only superior learning efficiency but also enhanced training stability.

**Ablations.** We carry out ablation studies to test the respective contributions of contrastive learning and attention. We compare ACORM to four ablations: i) `ACORM_w/o_CL`, it only excludes contrastive learning; ii) `ACORM_w/o_MHA`, it only removes the attention module; iii) `ACORM_w/o_MHA (Vanilla)`, it removes both the attention and state encoding, and directly feeds the current state into the mixing network like QMIX; and iv) `QMIX`, it removes all components. For ablations, all other structural modules are kept consistent strictly with the full ACORM.

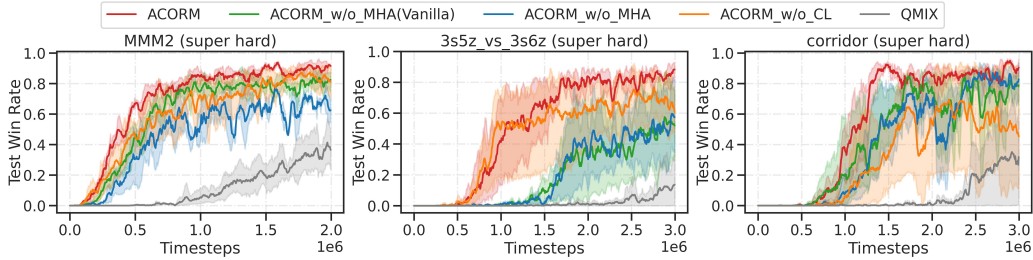

Figure 3: Ablation studies. ACORM_w/o_CL removes contrastive learning, ACORM_w/o_MHA removes attention, and ACORM_w/o_MHA (Vanilla) removes attention and state encoding.

Figure 3 shows ablation results on three super hard maps, and more ablations on other maps can be found in Appendix F. When either of the two components is removed, ACORM obtains decreased performance and still outperforms QMIX. It demonstrates that both components are essential for ACORM's capability and they are complementary to each other. Especially, both ACORM_w/o_CL and ACORM_w/o_MHA achieve significant performance gains compared to QMIX, which further verifies the respective effectiveness in tackling complex tasks. Specifically, ACORM_w/o_MHA (Vanilla) obtains very similar performance compared to ACORM_w/o_MHA, indicating that the effectiveness comes from the attention module other than encoding the state trajectory via a GRU.

## 3.2 CONTRASTIVE ROLE REPRESENTATIONS

To answer the second question, we gain deep insights into learned role representations through visualization on the example MMM2 task, where the agent controls a team of units (1 Medivac, 2 Marauders, and 7 Marines) to battle against an opposing army (1 Medivac, 3 Marauders, and 8 Marines). Fig. 4 presents example rendering scenes in an evaluation trajectory of the trained ACORM policy. Initially ($t = 1, 12$), all agent embeddings tend to be crowded together with limited discrimination, and the K-means algorithm moderately separates them into several clusters. Via contrastive learning, the acquired role representations within the same cluster are pushed closer to each other, and those in different clusters are notably separated. At a later stage ($t = 40$), agent embeddings are already scattered widely throughout the space with a good clustering effect so far. This phenomenon indicates that the system has learned effective role assignment with heterogeneous behavior patterns. Then, the role encoder transforms these agent embeddings into more discriminative role representations.

The team composition naturally evolves over time. At $t = 1$, Marauders $\{0, 1\}$ form a group and Marines $\{2, 3, 4, 5, 6, 7\}$ form another due to their intrinsic agent heterogeneity. In the middle of the battle $t = 12$, Marauders $\{0, 1\}$ join the same group of Marines $\{2, 4, 7, 6, 8\}$ to focus fire on enemies, while Marines $\{3, 5\}$ separate from the offense team since they are severely injured. Late in the battle at $t = 40$, Marines $\{2, 3, 4, 6, 7\}$ are still in the offense team, while Marauders $\{0, 1\}$ and Marine 5 fall into the same dead group. In summary, it is clearly verified that ACROM learns meaningful role representations associated with agent's behavior patterns and achieves effective dynamic team composition. More insights and explanations can be found in Appendix G.

## 3.3 ATTENTION-GUIDED ROLE COORDINATION

To answer the last question, we visualize attention weights $\boldsymbol{\alpha}$ in Eq. (6) using heatmaps, as shown in Fig. 5. The number of agent clusters is $K = 4$. In most heads, roles in the same cluster have similar attention weights, while different clusters exhibit significantly varying weights (e.g., in all four heads at $t = 10$, the weight distribution over four clusters: $\{0, 1, 2, 3\}, \{5\}, \{4, 6, 7, 8\}, \{9\}$). This phenomenon indicates that the global state has successfully attended to the learned role patterns.

The attention mechanism draws several interesting insights on the battle, such as: i) Head 2 evidently attends to the injury-rescue pattern, since the largest weights come from Medivac 9 and low-health units (Marauder 1 at $t = 4$, Marine 5 at $t = 10$, and Marines $\{3, 4, 5, 6, 7\}$ at $t = 36$). ii) In most heads, attention weights of Marauders $\{0, 1\}$ are usually high at the beginning, and are significantly decreased over time. It corresponds to the behavior pattern that Marauders play an important role

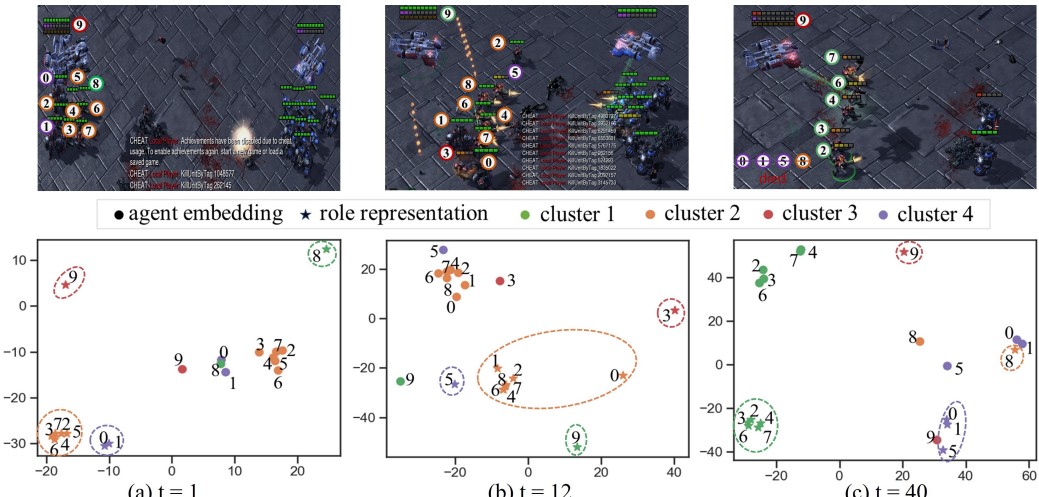

Figure 4: Example rendering scenes at three time steps in an evaluation trajectory generated by the trained ACORM policy on `MMM2`. The upper row shows screenshots of combat scenarios that contain the information of positions, health points, shield points, states of ally and enemy units, etc. The lower row visualizes the corresponding agent embeddings (denoted with bullets '•') and role representations (denoted with stars '⋆') by projecting these vectors into 2D space via t-SNE for qualitative analysis, where agents within the same cluster are depicted using the same color.

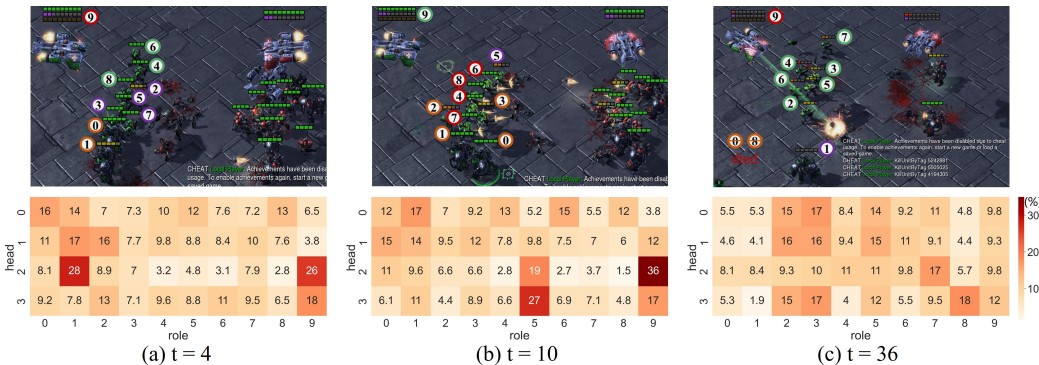

Figure 5: Example rendering scenes in an evaluation trajectory generated by the trained ACORM policy on `MMM2`. The lower row visualizes attention weights ($\alpha$ in Eq. (6)) of all four heads that explain how the global state attends to each role to guide skillful coordination in the role space. A higher weight means a larger contribution made by the corresponding role for value decomposition.

at early attacks and the offensive mission will gradually be handed over to Marines. iii) On the verge of victory at $t = 36$, the primary concern is using Marines $\{2, 3, 5\}$ to make final attacks with low-health Marines $\{4, 6, 7\}$ providing auxiliary support. Obviously, heads $\{0, 1\}$ intuitively reflect this strategy, as Marines $\{2, 3, 5\}$ have the highest weights, followed by Marines $\{4, 6, 7\}$ and all other units. Moreover, the capability of our attention module could be much more profound than these examples from the superficial visualization.

## 4 RELATED WORK

**Agent Heterogeneity.** As a compelling paradigm, CTDE (Foerster et al., 2016) has yielded numerous algorithms (Lowe et al., 2017; Son et al., 2019; Wang et al., 2023). Many of them share policy parameters to improve learning efficiency and scale to large-scale systems, which results in homogeneous behaviors across agents (Liu et al., 2022). To promote diversity, SePS (Christianos et al.,

2021) partitions agents into a fixed set of groups and shares parameters within the same group only, while ignoring evolving dynamics of the team. GoMARL (Zang et al., 2023) generates dynamic groups with an automatic grouping mechanism to possess diverse strategies. MAVEN (Mahajan et al., 2019) learns diverse exploratory behaviors by introducing a latent space for hierarchical control. CDS (Li et al., 2021) equips each agent with an additional local Q-function to decompose the policy to the shared and non-shared part. EOI (Jiang & Lu, 2021) promotes individuality by encouraging agents to visit their own familiar observations. CIA (Liu et al., 2023) boosts agent distinguishability in value decomposition via contrastive learning. While differentiating each agent from the rest, these methods neglect the development of effective team composition with implicit task allocation, and might hinder the discovery of sophisticated coordination patterns.

**Role Emergence.** Researchers have also introduced the role concept into multi-agent tasks (Sims et al., 2008; Lhaksmana et al., 2018; Xia et al., 2023; Cao et al., 2023), or similarly, the concept of skills (Yang et al., 2020a) or subtasks (Yuan et al., 2022). ROMA (Wang et al., 2020) conditions individual policies on roles and solely relies on the current observation to generate the role embedding, which might be inadequate for capturing complex agent behaviors. RODE (Wang et al., 2021) associates each role with a fixed subset of the full action space to reduce learning complexity. Following RODE, SIRD (Zeng et al., 2023) transforms role discovery into hierarchical action space clustering. Nonetheless, they neglect the evolving dynamics of the team since roles are kept fixed in the training stage. For dynamic role assignment, some works learn identity representations to group agents during training, and maintain a selection strategy to realize the assignment from agents to skills (Liu et al., 2022) or subtasks (Yang et al., 2022; Iqbal et al., 2022). Nevertheless, they encode the identity solely from a one-hot vector, which might be insufficient to distinguish complex agent characteristics. COPA (Liu et al., 2021) realizes dynamic role allocation via periodically distributing a global view of team composition to each agent even in execution. However, it relaxes the CTDE constraint by introducing communication during decentralized execution, and the global composition is simply sampled from a fixed set of teams.

In summary, our method differs from the above approaches involving the role concept, and exhibits several promising advantages. Our method strictly follows the CTDE paradigm, accommodates the dynamic nature of multi-agent systems, and learns more efficient role representations.

**Contrastive Learning and Attention Mechanism**. Contrastive learning is gaining widespread popularity for self-supervised representation learning in various domains (He et al., 2020; Su et al., 2022; Laskin et al., 2020). As a simple and effective technique, contrastive learning is also investigated to assist MARL tasks, such as boosting the credit-level distinguishability (Liu et al., 2023), facilitating the utilization of the agent-level contextual information (Song et al., 2023), and grounding agent communication (Lo & Sengupta, 2022). In this study, we apply contrastive learning to optimize role representations, facilitating sophisticated coordination with better role assignment.

Attention is the fundamental building block of famous transformer architectures (Vaswani et al., 2017) that exhibit growing dominance in advances of AI research (Brown et al., 2020; Dosovitskiy et al., 2021; Chen et al., 2021b). Due to its superiority in extracting dependencies between sequences, attention has been widely applied to MARL domains for various utilities, such as learning a centralized critic (Iqbal & Sha, 2019), concentrating on relevant subtasks (Yuan et al., 2022), formulating MARL as a sequence modeling problem (Wen et al., 2022), addressing stochastic partial observability (Phan et al., 2023), etc (Yang et al., 2020b; Shao et al., 2023; Zhai et al., 2023). In this study, we use attention to guide skillful role coordination for more expressive credit assignment.

## 5 CONCLUSION AND DISCUSSION

In this paper, we propose a general framework that learns contrastive role representations to promote behavior heterogeneity and knowledge transfer across agents, and facilitates skillful coordination in a sophisticated role space via an attention mechanism. Experimental results and ablations verify the superiority of our method, and deep insights via visualization demonstrate the achievement of meaningful role representations and skillful role coordination. Though, our method does not consider the exploitation of history exploratory trajectories for extracting roles, and also needs an explicit clustering component with a pre-defined number of total behavior patterns. We leave these directions as future work. Moreover, extending our framework to offline settings is a promising line for practical scenarios where online interaction is expensive and even infeasible.

ACKNOWLEDGEMENTS

The work was supported by the National Natural Science Foundation of China under Grant 62376122, Grant 62073160, Grant 62276126, and Grant 62176116.

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

APPENDIX A. CONTRASTIVE ROLE REPRESENTATION LEARNING

In this section, we give the proof of Theorem 1 in the text based on introducing a lemma as follows.

**Lemma 1.** *Given a role from the distribution $M \sim P(M)$, let $e = f_\phi(\sum_t(o^t, a^{t-1}))$ as the agent embedding generated by role $M$, and $z \sim f_\theta(z|e)$, where $\sum_t(o^t, a^{t-1})$ is the agent's local trajectory following a given policy. Then, we have*

$$\frac{p(M|z)}{p(M)} = \mathbb{E}_e\left[\frac{p(z|e)}{p(z)}\right]. \tag{9}$$

*Proof.*

$$\begin{aligned}
\frac{p(M|z)}{p(M)} &= \frac{p(z|M)}{p(z)} \\
&= \int_e \frac{p(e|M)p(z|e)}{p(z)}\mathrm{d}e \\
&= \mathbb{E}_e\left[\frac{p(z|e)}{p(z)}\right].
\end{aligned} \tag{10}$$

The proof is completed. □

**Theorem 1.** *Let $\mathcal{M}$ denote a set of roles following the role distribution $P(M)$, and $|\mathcal{M}| = K$. $M \in \mathcal{M}$ is a given role. Let $e = f_\phi(\sum_t(o^t, a^{t-1}))$, $z \sim f_\theta(z|e)$, and $h(e, z) = \frac{p(z|e)}{p(z)}$, where $\sum_t(o^t, a^{t-1})$ is the agent's local trajectory following a given policy. For any role $M^* \in \mathcal{M}$, let $e^*$ denote the agent embedding generated by the role $M^*$, then we have*

$$I(z; M) \geq \log K + \mathbb{E}_{\mathcal{M},z,e}\left[\log \frac{h(e, z)}{\sum_{M^* \in \mathcal{M}} h(e^*, z)}\right]. \tag{2}$$

*Proof.* Using Lemma A.1 and Jensen's inequality, we have

$$\begin{aligned}
\mathbb{E}_{\mathcal{M},z,e}\left[\log \frac{h(e, z)}{\sum_{M^* \in \mathcal{M}} h(e^*, z)}\right] &= \mathbb{E}_{\mathcal{M},z,e}\left[\log \frac{\frac{p(z|e)}{p(z)}}{\frac{p(z|e)}{p(z)} + \sum_{M^* \in \mathcal{M}\setminus M}\frac{p(z|e^*)}{p(z)}}\right] \\
&= \mathbb{E}_{\mathcal{M},z,e}\left[-\log\left(1 + \frac{p(z)}{p(z|e)}\sum_{M^* \in \mathcal{M}\setminus M}\frac{p(z|e^*)}{p(z)}\right)\right] \\
&\approx \mathbb{E}_{\mathcal{M},z,e}\left[-\log\left(1 + \frac{p(z)}{p(z|e)}(K-1)\mathbb{E}_{M^* \in \mathcal{M}\setminus M}\left[\frac{p(z|e^*)}{p(z)}\right]\right)\right] \\
&= \mathbb{E}_{\mathcal{M},z,e}\left[-\log\left(1 + \frac{p(z)}{p(z|e)}(K-1)\right)\right] \\
&= \mathbb{E}_{\mathcal{M},z,e}\left[\log\left(\frac{1}{1 + \frac{p(z)}{p(z|e)}(K-1)}\right)\right] \\
&\leq \mathbb{E}_{\mathcal{M},z,e}\left[\log\left(\frac{1}{\frac{p(z)}{p(z|e)}K}\right)\right] \\
&\leq \mathbb{E}_{\mathcal{M},z}\left[\log \mathbb{E}_e\left[\frac{p(z|e)}{p(z)}\right]\right] - \log K \\
&= \mathbb{E}_{\mathcal{M},z}\left[\log \frac{p(M|z)}{p(M)}\right] - \log K \\
&= I(z; M) - \log K.
\end{aligned}$$

$$\tag{11}$$

Thus, we complete the proof. □

# APPENDIX B. IMPLEMENTATION DETAILS AND EXTENDED EXPERIMENTS OF QMIX-BASED ACORM

Based on the implementations in Section 2, we summarize the brief procedure of ACORM based on QMIX in Algorithm 1.

---

**Algorithm 1:** ACORM based on QMIX

---

**Input:** $\phi$: agent's trajectory encoder
   $\theta$: role encoder
   $K$: number of clusters
   $T_{cl}$: time interval for updating contrastive loss
   $n$: number of agents
   $\mathcal{B}$: replay buffer
   $T$: time horizon of a learning episode

1 Initialize all network parameters
2 Initialize the replay buffer $\mathcal{B}$ for storing agent trajectories
3 **for** *episode* $= 1, 2, ...$ **do**
4   Initialize history agent embedding $e_i^0$, and action vector $a_i^0$ for each agent
5   **for** $t = 1, 2, ..., T$ **do**
6     Obtain each agent's partial observation the $\{o_i^t\}_{i=1}^n$ and global state $\boldsymbol{s}^t$
7     **for** *agent* $i = 1, 2, ..., n$ **do**
8       Calculate the agent embedding $e_i^t = f_\phi(o_i^t, a_i^{t-1}, e_i^{t-1})$
9       Calculate the role representation $z_i^t = f_\theta(e_i^t)$
10      Select the local action $a_i^t$ according to individual Q-function $Q_i(e_i, a_i^t)$
11     **end**
12     Execute joint action $\boldsymbol{a}^t = [a_1^t, a_2^t, ..., a_n^t]^\top$, and obtain global reward $r^t$
13   **end**
14   Store the trajectory to $\mathcal{B}$
15   Sample a batch of trajectories from $\mathcal{B}$
16   **if** *episode* mod $T_{cl} == 0$ **then**
17     Partition agent embeddings $\{e_i^t\}_{i=1}^n$ into $K$ clusters $\{C_j\}_{j=1}^K$ using K-means
18     **for** *agent* $i = 1, 2, ..., n$ **do**
19       Construct positive keys $\{z_{i'}\}_{i' \in C_j}$ and negative keys $\{z_{i*}\}_{i* \notin C_j}$ for query $z_i, i \in C_j$
20     **end**
21     Update contrastive learning loss according to Eq. (4)
22     Update momentum role encoder according to Eq. (5)
23   Calculate attention output $\boldsymbol{\tau}_{\text{mha}}$ via prompting the global state to attend to role representations $\{z_i\}_{i=1}^n$ in Eqs. (6)-(8)
24   Concatenate $\boldsymbol{\tau}_{\text{mha}}$ with state embedding $\boldsymbol{\tau}$ to form the input to the mixing network
25   Update the parameters of individual Q-network and the mixing network
26 **end**

---

In this paper, we use simple network structures for the trajectory encoder, the role encoder, and the attention mechanism. Specifically, the trajectory encoder contains a fully-connected multi-layer perceptron (MLP) and a GRU network with ReLU as the activation function, and encodes agent's trajectory into a 128-dimensional embedding vector. The role encoder is a fully-connected MLP that transforms the 128-dimensional agent embedding into a 64-dimensional role representation. The setting of the mixing network is kept as the same as that of QMIX (Rashid et al., 2020), where the architecture contains two 32-dimensional hidden layers with ReLU activation. Table 1 shows the details of network structures.

For all tested algorithms, we use the Adam optimizer with a learning rate of 6e-4. For exploration, we use the $\epsilon$-greedy strategy with $\epsilon$ annealed linearly from 1.0 to 0.02 over $80k$ time steps and kept constant for the rest of the training. Every time an entire episode from online interaction is collected and stored in the buffer, the Q-networks are updated using a batch of 32 episodes sampled from the replay buffer with a capacity of 5000 state transitions. The target Q-network is updated using a soft update strategy with momentum coefficient 0.005. The contrastive learning loss is jointly trained every 100 steps of the Q-network updates. The decentralized policy is evaluated every $5k$ update steps with 32 episodes generated. For all domains, the number of clusters in ACORM is set to $K = 3$. Appendix D provides an analysis on this hyperparameter, and the experiments show that the performance of ACORM is not significantly affected by the value of $K$. The details of hyperparameters can be found in Table 2.

Table 1: The network configurations used for ACORM based on QMIX.

| Network Configurations | Value | Network Configurations | Value |
|---|---|---|---|
| role representation dim | 64 | hypernetwork hidden dim | 32 |
| agent embedding dim | 128 | hypernetwork layers num | 2 |
| state embedding dim | 64 | type of optimizer | Adam |
| attention output dim | 64 | activation function | ReLU |
| attention head num | 4 | add last action | True |
| attention embedding dim | 128 | | |

Table 2: Hyperparameters used for ACORM based on QMIX.

| Hyperparameter | Value | Hyperparameter | Value |
|---|---|---|---|
| buffer size | 5000 | start epsilon $\epsilon_s$ | 1.0 |
| batch size | 32 | finish epsilon $\epsilon_f$ | 0.02 |
| learning rate | $6 \times 10^{-4}$ | $\epsilon$ decay steps | 80000 |
| use learning rate decay | True | evaluate interval | 5000 |
| contrastive learning rate | $8 \times 10^{-4}$ | evaluate times | 32 |
| momentum coefficient $\beta$ | 0.005 | target update interval | 200 |
| update contrastive loss interval $T_{cl}$ | 100 | discount factor $\gamma$ | 0.99 |
| cluster num | 3 | | |

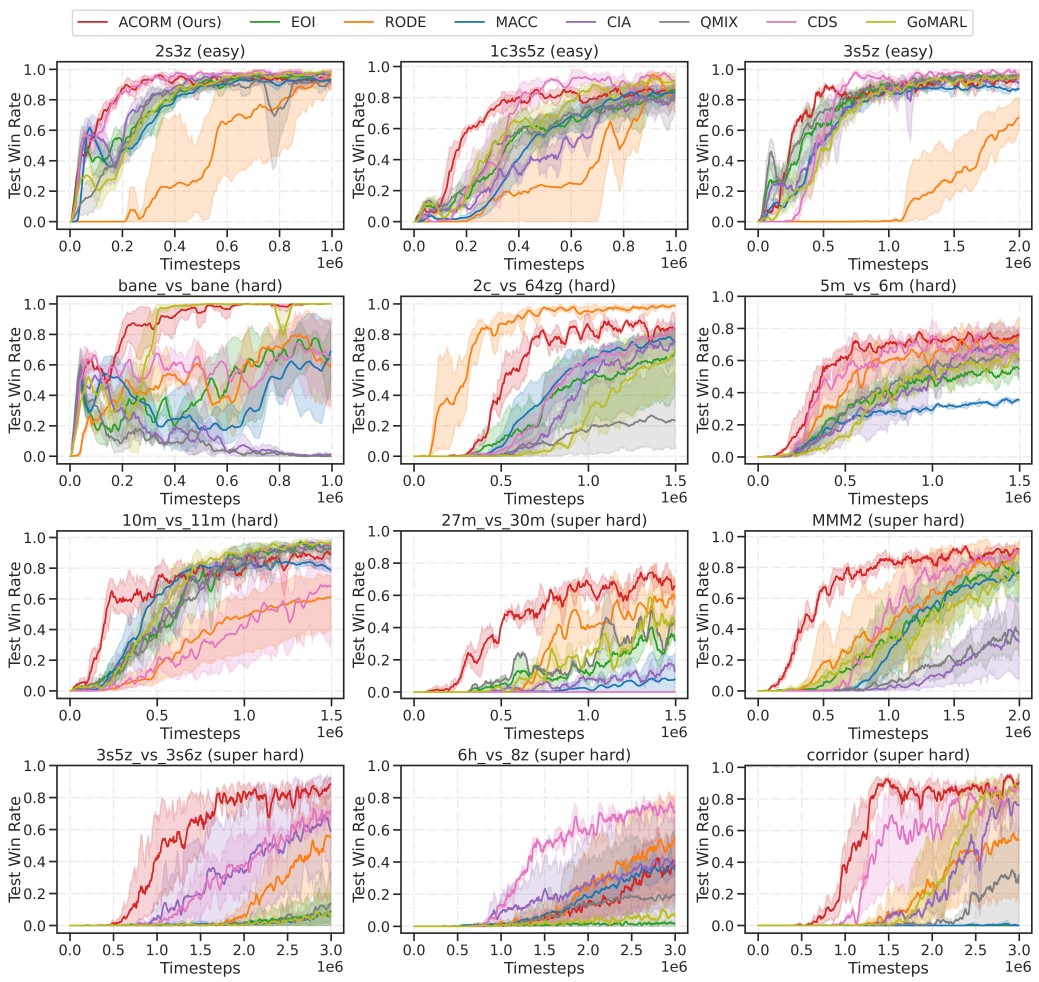

Figure 6: Extended performance comparison between ACORM and baselines on 12 SMAC maps.

Fig. 6 presents the extended performance of ACORM on 12 SMAC maps. Obviously, the observations and conclusions from the extended performance are kept consistent with those in Sec. 3.1 of the main paper.

## APPENDIX C. EXTENDED EXPERIMENTS ON GOOGLE RESEARCH FOOTBALL

In addition to SMAC environments, we also benchmark our approach on three challenging Google research football (GRF) offensive scenarios as

- `academy_3_vs_1_with_keeper`: Three of our players try to score from the edge of the box, one on each side, and the other at the center. Initially, the player at the center has the ball and is facing the defender. There is an opponent keeper.

- `academy_counterattack_hard`: 4 versus 2 counter-attack with keeper; all the remaining players of both teams run back towards the ball.

- `academy_run_to_score_with_keeper`: Our player starts in the middle of the field with the ball, and needs to score against a keeper. Five opponent players chase ours from behind.

In GRF tasks, agents need to coordinate timing and positions for organizing offense to seize fleeting opportunities, and only scoring leads to rewards. In our experiments, we control left-side players except the goalkeeper. The right-side players are rule-based bots controlled by the game engine. Agents have a discrete action space of 19, including moving in eight directions, sliding, shooting, and passing. The observation contains the positions and moving directions of the ego-agent, other agents, and the ball. The $z$-coordinate of the ball is also included. Environmental reward only occurs at the end of the game. They will get $+100$ if they win, else get $-1$.

Fig. 7 presents the performance comparison between ACORM and baselines on three challenging GRF scenarios. It can be observed that QMIX obtains poor performance, since the tasks in GRF are more challenging than in the SMAC benchmark. In contrast, ACORM gains a significantly improved increase in the test win rate, especially in the first 1M training timesteps, which successfully demonstrates the effectiveness of our method evaluated on GRF benchmarks. Together with evaluations on SMAC domains, the same conclusion can still be drawn that ACORM outperforms baseline methods by a larger margin on harder tasks that demand a significantly higher degree of behavior diversity and coordination. In summary, experimental results on extended GRF environments are generally consistent with those on the SMAC benchmark. .

Due to the very limited time for rebuttal revision, we only compare ACORM to QMIX and CDS, as other baselines (RODE, EOI, MACC, CIA) are not evaluated on GRF in their original papers. Also, we just show the performance within 2M training steps. We hope that the extended experimental evaluation could demonstrate adequate persuasiveness of our method. We are rushing to conduct the evaluation of baseline methods on GRF scenarios with more training timesteps, and trying to update them before the rebuttal deadline.

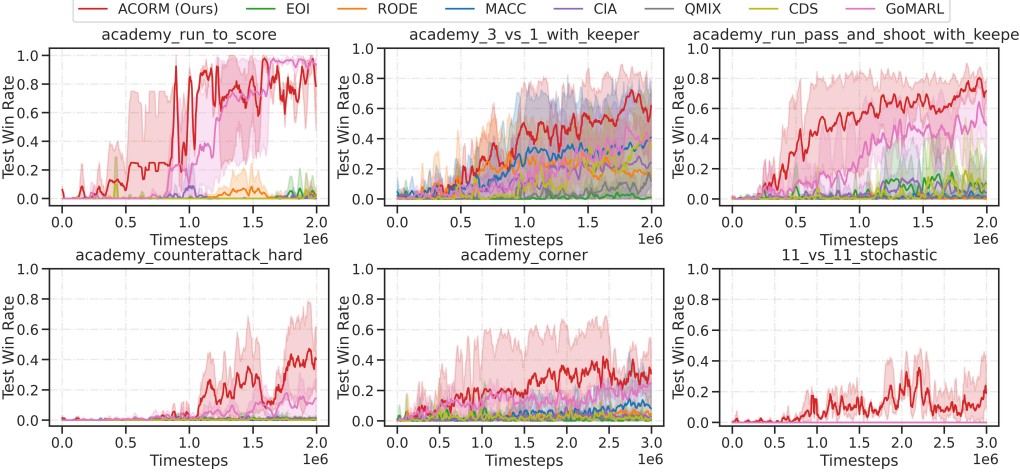

Figure 7: Performance comparison between ACORM and baselines on three GRF scenarios.

## APPENDIX D. ACORM BASED ON MAPPO

In addition, we realize ACORM on top of the MAPPO algorithm, as show in Fig. 8. Most of the network structure is kept the same as QMIX-based ACORM, including the agent embedding, the role encoder, and the attention mechanism. To align with MAPPO that uses an actor-critic architecture, we input both the agent's observation and the augmented global state $\tilde{s}$ (obtained from the attention mechanism in Fig. 8(e)) into the critic, as shown in Fig. 8(a). Tables 3 and 4 present the detailed network structure and experimental hyperparameters, respectively.

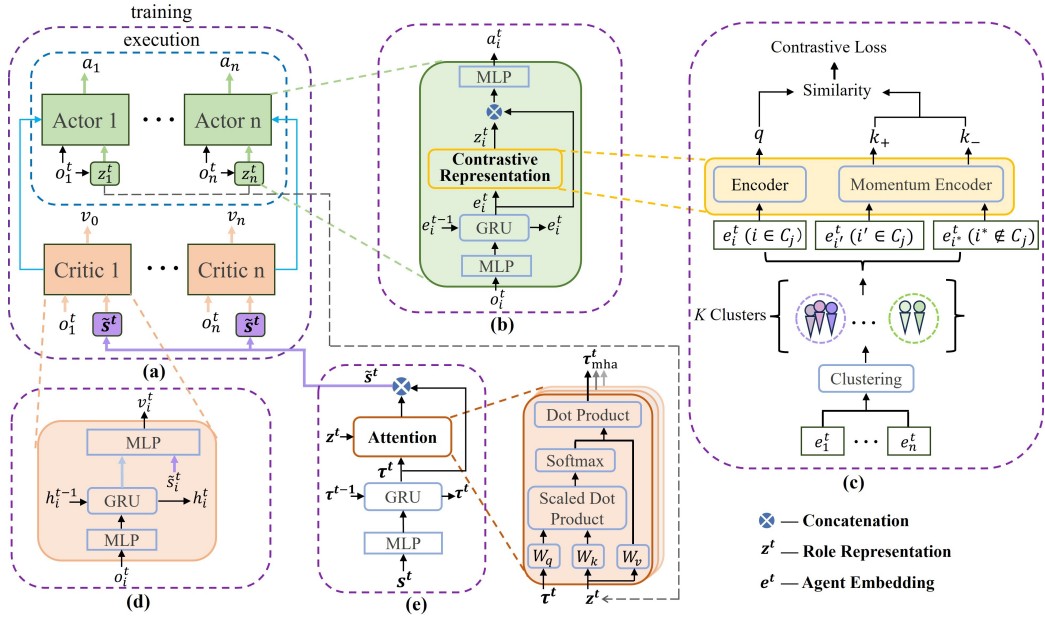

Figure 8: The ACORM framework based on MAPPO. (a) The overall architecture. (b) The shared actor network structure for each agent, where the role representation is extracted from agent's trajectory. (c) The detail of learning role representations via contrasting learning. (d) The shared critic network structure for each agent. (e) The attention module that incorporates learned role representations into value decomposition.

Table 3: The network configurations used for ACORM based on MAPPO.

| Network Configurations | Value | Network Configurations | Value |
|---|---|---|---|
| role representation dim | 64 | attention output dim | 64 |
| agent embedding dim | 128 | attention head num | 4 |
| state embedding dim | 64 | attention embedding dim | 128 |
| critic RNN hidden dim | 64 | type of optimizer | Adam |
| add agent ID | False | activation | ReLU |

Table 4: Hyperparameters used for ACORM based on MAPPO.

| Hyperparameter | Value | Hyperparameter | Value |
|---|---:|---|---:|
| batch size | 32 | entropy coefficient | 0.02 |
| mini batch size | 32 | cluster num | 3 |
| actor learning rate | $6 \times 10^{-4}$ | discount factor $\gamma$ | 0.99 |
| critic learning rate | $8 \times 10^{-4}$ | momentum coefficient $\beta$ | 0.005 |
| contrastive learning rate | $8 \times 10^{-4}$ | evaluate interval | 5000 |
| update contrastive loss interval $T_{cl}$ | 32 | evaluate times | 32 |
| clip | 0.2 | use advantage normalization | True |
| GAE lambda $\lambda$ | 0.95 | use learning rate decay | False |
| K epochs | 5 | | |

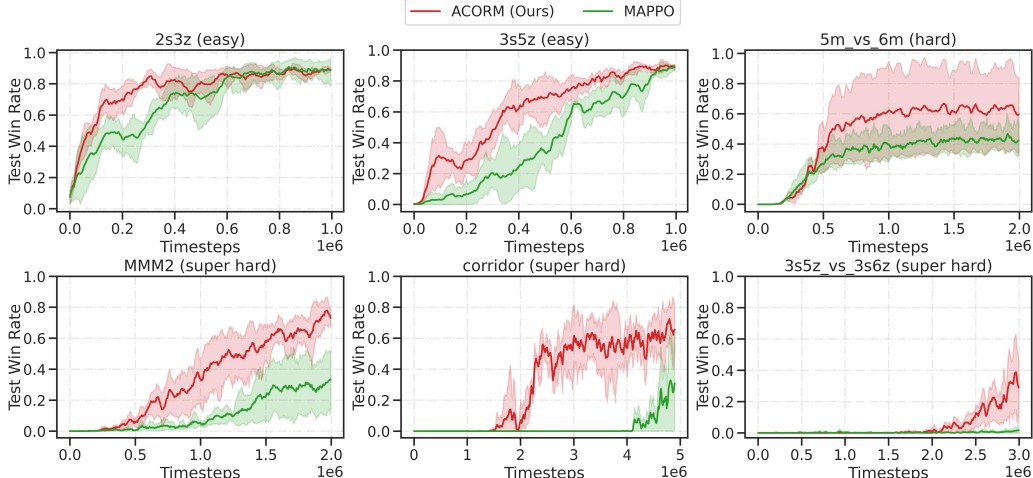

Figure 9: Performance comparison between MAPPO-based ACORM and the MAPPO baseline on representative maps, including two easy levels (2s3z, 3s5z), one hard level (5m_vs_6m), and three super hard levels (MMM2, corridor, 3s5z_vs_3s6z).

Figure 9 presents the performance of MAPPO-based ACORM on six representative tasks: 2s3z, 3s5z, 5m_vs_6m, MMM2, corridor, and 3s5z_vs_3s6z. It can be observed that ACORM achieves a significant performance improvement over MAPPO. Akin to the QMIX-based version, ACORM outperforms MAPPO by the largest margin on super hard maps that demand a significantly higher degree of behavior diversity and coordination. Again, experimental results demonstrate ACORM's superiority of learning efficiency in complex multi-agent domains.

## APPENDIX E. HYPERPARAMETER ANALYSIS ON THE NUMBER OF CLUSTERS

We test the influence of the number of clusters $K$ on ACROM's performance. Fig. 10 shows the performance of ACORM with varying values of $K = 2, 3, 4, 5$. Generally speaking, ACORM obtains similar performance across different values of $K$. It demonstrates that ACORM achieves good learning stability and robustness as the performance is insensitive to the pre-defined number of role clusters. An outlier case is observed in the 5m_vs_6m map where the ACORM's performance drops a little when $K = 5$. This is likely because there are only 5 agents in 5m_vs_6m. When $K = 5$, each agent represents a distinct role cluster. It forces the strategies of each agent to diverge, which might not be conductive to realize effective team composition across agents.

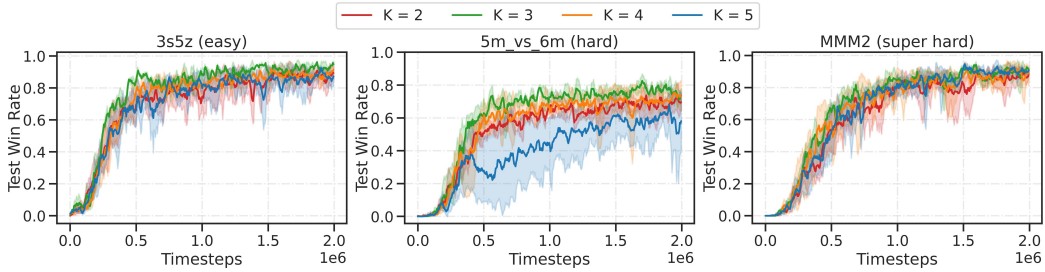

Figure 10: Hyperparameter analysis on the number of clusters $K$ in negative pairs generation.

To illustrate why ACORM performs well across different values of $K$, we show visualizations from different learned ACORM policies with $K = 3, 4, 5$ in Fig. 11. When the clustering granularity is coarse as $K = 3$ in Fig. 11(a), even within the same cluster, ACORM can still learn meaningful role representations with distinguishable patterns. Agent embeddings of Marines $\{2, 3, 4, 5, 6, 7, 8\}$ are crowded together with limited discrimination. Via contrastive learning, the obtained role representations exhibit an interesting self-organized structure where Marines $\{2, 5, 6\}$ and $\{3, 4, 7, 8\}$ implicitly form two distinctive sub-groups. It can be observed from the rendering scene that Marines $\{2, 5, 6\}$ and $\{3, 4, 7, 8\}$ are all at an attacking stage, while the former sub-group is in lower health than the latter. On the other hand, with a fine granularity of $K = 5$ in Fig. 11(c), the contrastive learning module transforms clustered agent embeddings into more discriminative role representations. For example, while Marines $2, 3, 5, 6, 8$, and $4, 7$ form three clusters, their role representations are still closer to each other and farther from Marauders $\{0, 1\}$ and Medivac $\{9\}$, since they are the same type of agents with similar behavior patterns. In summary, it again demonstrates that ACORM learns meaningful role representations and achieves effective and robust team composition.

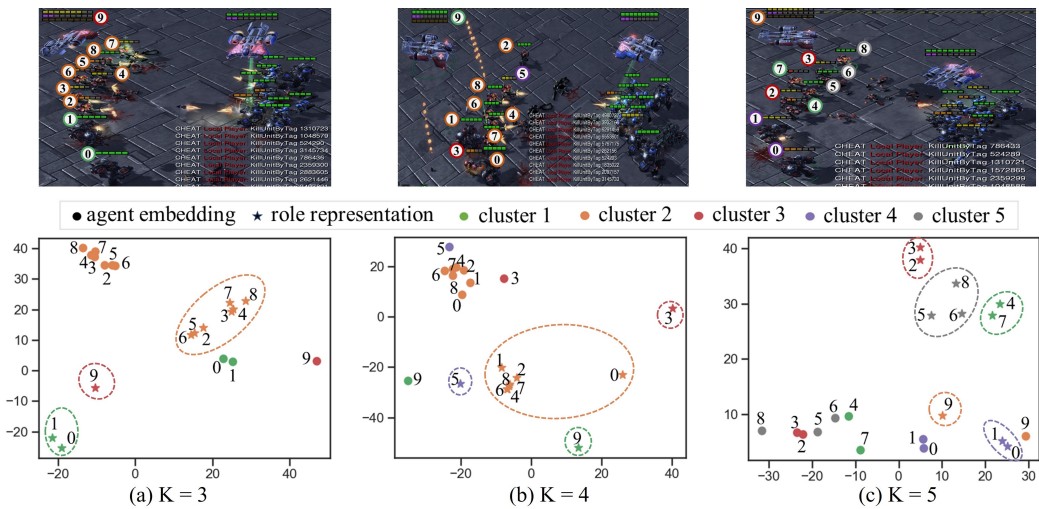

Figure 11: Visualization with different number of clusters $K$ in negative pairs generation.

## APPENDIX F. MORE ABLATION RESULTS

Figure 12 presents the full ablation results of ACORM on all six representative maps. It again demonstrates the significance of both the contrastive learning module and the attention mechanism for ACORM's performance. A noteworthy point is that both ACORM_w/o_CL and ACORM_w/o_MHA gain remarkable performance improvement by the largest margin on super hard maps, which further validate ACORM's advantages of promoting diversified behaviors and skillful coordination in complex multi-agent tasks. Another interesting observation is that when omitting the contrastive learning module, there is a notable increase in the variance of learning curves. It can be interpreted as an evidence that contrastive learning helps training more robust role representations and enhances learning stability.

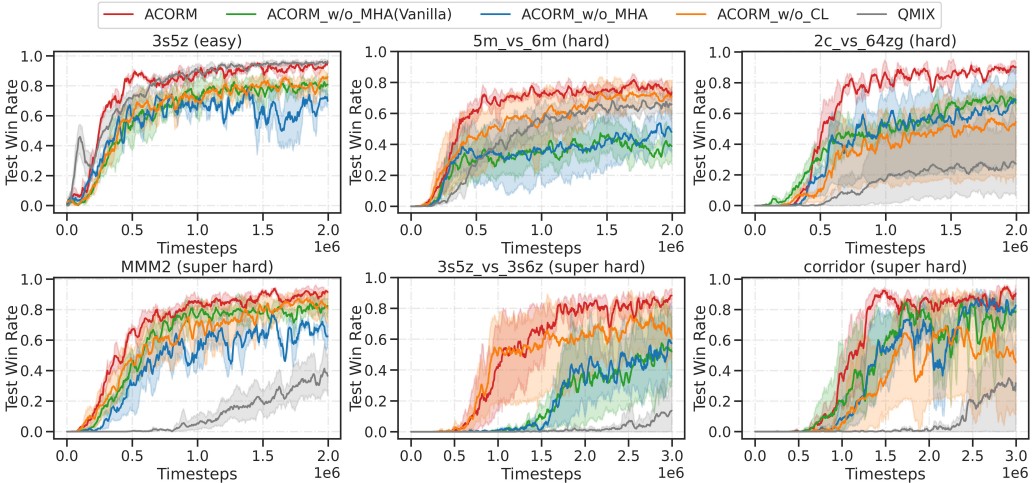

Figure 12: Full ablation results on ACORM. ACORM_w/o_CL means removing contrastive learning, ACORM_w/o_MHA represents excluding attention, ACORM_w/o_MHA (Vanilla) represents excluding attention and state encoding, and QMIX corresponds to removing all components.

## APPENDIX G. MORE INSIGHTS ON CONTRASTIVE REPRESENTATION LEARNING

Based on Section 3.2, here we provide more insights on the effectiveness of learned role representations. Indeed, one reason for ACORM's significant improvement on learning efficiency comes from its capability of facilitating implicit knowledge transfer across similar agents throughout the entire learning process. For example, Marines $\{2, 3, 4, 5, 6, 7\}$ form a group at $t = 1$, Marauders $\{0, 1\}$ and Marines $\{2, 4, 6, 7, 8\}$ form a group at $t = 12$, and Marines $\{2, 3, 4, 6, 7\}$ form a group at $t = 40$. It can be observed from the rendering scenes that these three groups are all responsible for attacking enemies. At different time steps, agents in the attacking group can share similar role representations to promote knowledge transfer, even if they belong to heterogeneous agent types. This implicit transfer across agents and across timesteps can significantly increase the exploration efficiency of agents.

Another highlight of ACORM is the promotion of behavior heterogeneity, even if agents have the same innate characteristics. For example, while Marines $\{2, 3, 4, 5, 6, 7, 8\}$ belong to the same agent type, they are distributed to different groups with heterogeneous roles as: 1) at $t = 12$, Marines $\{2, 4, 6, 7, 8\}$ with role of attacking, and Marines $\{3\}$ and $\{3\}$ with role of the wounded; and 2) at $t = 40$, Marines $\{2, 3, 4, 6, 7\}$ with role of attacking, and Marines $\{5\}$ and $\{8\}$ with role of the dead. In general, even though some information is lost during dimension reduction using t-SNE, it is evident that our role representation still manages to exhibit such remarkable results.

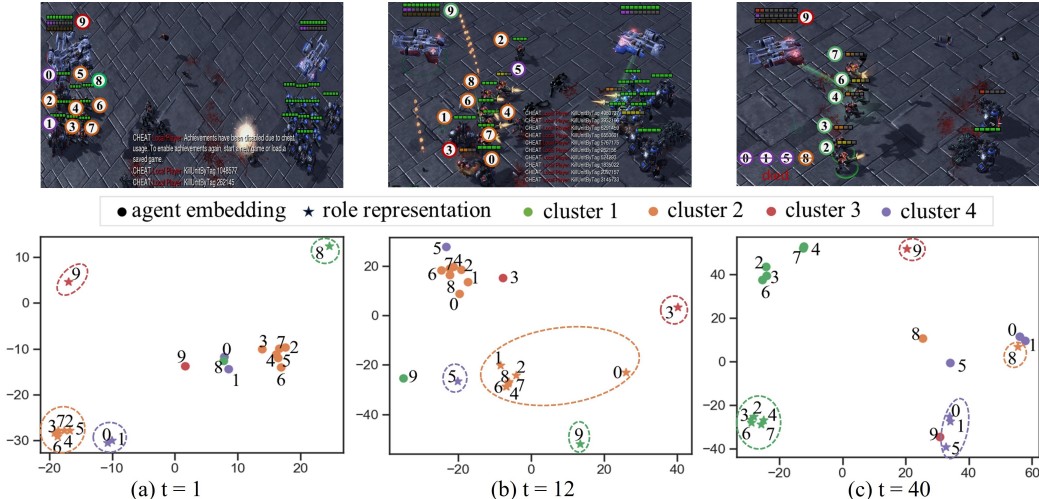

Figure 13: Example rendering scenes at three time steps in an evaluation trajectory generated by the trained ACORM policy on MMM2. The upper row shows screenshots of combat scenarios that contain the information of positions, health points, shield points, states of ally and enemy units, etc. The lower row visualizes the corresponding agent embeddings (denoted with bullets '●') and role representations (denoted with stars '★') by projecting these vectors into 2D space via t-SNE for qualitative analysis, where agents within the same cluster are depicted using the same color.

