# OpenReview forum: "Attention-Guided Contrastive Role Representations for Multi-agent Reinforcement Learning"
_ICLR.cc/2024/Conference — ICLR 2024 poster_

### Official Review · Reviewer_jtxX · 2023-10-25

**Soundness:** 3 good
**Presentation:** 3 good
**Contribution:** 3 good
**Rating:** 6
**Confidence:** 4

**Summary:**

The paper proposes a method of attention-guided promotion (incorporating attention mechanisms in the global state to participate in value decomposition) to maximize mutual information to formalize role representation and derive a contrastive learning objective function. ACORM choose the StarCraft multi-agent challenge (SMAC) benchmark and achieves state-of-the-art performance on most hard and superhard maps.

**Strengths:**

Using mutual information to distinguish roles

The experiment was conducted on a difficult map in StarCraft

The experimental diagram is very detailed

**Weaknesses:**

There is no reasonable explanation or formula for the promotion of credit assignment  by attention, and the paper only demonstrates the effectiveness of the method through experiments

**Questions:**

1. Does GRU encoding S play a more significant role in the effect ?

2. How do you know the state encoding after this attention, input to the mix network can have an impact on credit assignment ?

---

> ### Author Response · Authors · 2023-11-18
> **Response**
>
> **Q1: There is no reasonable explanation or formula for the promotion of credit assignment by attention, and the paper only demonstrates the effectiveness of the method through experiments.**
>
> A1: Thank you for your insightful comments.
> Throughout the paper, we have provided explanations for the promotion of more expressive credit assignment in many places.
> For example, the first paragraph of Sec. 2.2 (top, page 5) presents the line our research motivation for introducing the attention mechanism as "Popular CTDE algorithms, such as QMIX, realize behavior coordination across agents via a mixing network that estimates joint action-values as a monotonic combination of per-agent values, and the mixing network weights are conditioned on the system’s global state. Naturally, it is interesting to incorporate the learned role information into the mixing network to facilitate skillful coordination across roles. The simplest approach is to concatenate the global state and role representations for generating mixing network weights, while it fails to exploit the internal structure to effectively extract correlations in the role space. Fortunately, the attention mechanism aligns perfectly with our intention by prompting the global state to attend to learned role patterns, thus providing more expressive credit assignment in value decomposition."
> On the other hand, the visualization in Sec. 3.3 clearly illustrates that the global state has successfully attended to the learned role patterns, which in turn provides evidence for our attention-guided promotion of credit assignment.
>
> **Q2: Does GRU encoding $\bf{s}$ play a more significant role in the effect?**
>
> A2: We have included an additional ablation baseline "ACORM\_w/o\_MHA (Vanilla)" that removes both the attention module and the additional GRU-MLP structure.
> In this way, the input to the mixing network is only the current global state $\bf{s}$$^t$, which is kept the same as the original QMIX algorithm.
> In Sec. 3.1 of the updated paper and the updated Appendix F, it can be observed that ACORM\_w/o\_MHA (Vanilla) obtains very similar performance compared to ACORM\_w/o\_MHA.
> It can be concluded that the effectiveness of our attention module comes from prompting the global state to attend to role representations other than the GRU encoding of the global state trajectory.
>
>
> **Q3: How do you know the state encoding after this attention, input to the mix network can have an impact on credit assignment?**
>
> A3: As stated in A1, the idea of our attention-guided role coordination comes from QMIX's value decomposition.
> QMIX realizes behavior coordination across agents (i.e., credit assignment) via the mixing network whose weights are conditioned on the system’s global state.
> Intuitively, if we could condition the mixing network weights on extra role representations, we might realize more efficient role coordination for better credit assignment.
> Fortunately, the attention mechanism aligns perfectly with our intention by prompting the global state to attend to learned role patterns, thus providing more expressive credit assignment in value decomposition.
> Moreover, ablation studies (Sec. 3.1 and Appendix F) and visualization of role coordination (Sec. 3.3) have also verified that our attention module indeed has a positive impact on credit assignment.

---

> ### Author Response · Authors · 2023-11-21
> **Looking forward to further discussions!**
>
> Dear reviewer,
>
> We were wondering if our response and revision have resolved your concerns.
>
> In our responses, we have provided the source code and significantly extended the experimental evaluation. We conduct ablation studies with the same network size, and add another ablation baseline to **identify the respective effects of the attention module and the state trajectory encoding**. We have also **provided detailed explanations and insights on the attention-guided role coordination for more expressive credit assginment**.
>
> If our response has addressed your concerns, we would be grateful if you could re-evaluate our work.
>
> If you have any additional questions or suggestions, we would be happy to have further discussions.
>
> Best regards,
>
> The Authors

---

> ### Comment · Area_Chair_A6wJ · 2023-12-04
> **[Important] Response Required to Authors' Rebuttal**
>
> Dear Reviewer jtxX,
>
> As we progress through the review process for ICLR 2024, I would like to remind you of the importance of the rebuttal phase. The authors have submitted their rebuttals, and it is now imperative for you to engage in this critical aspect of the review process.
>
> Please ensure that you read the authors' responses carefully and provide a thoughtful and constructive follow-up. Your feedback is not only essential for the decision-making process but also invaluable for the authors.
>
> Thank you,
>
> ICLR 2024 Area Chair

---

### Official Review · Reviewer_1yPm · 2023-10-28

**Soundness:** 3 good
**Presentation:** 3 good
**Contribution:** 2 fair
**Rating:** 6
**Confidence:** 5

**Summary:**

This paper proposes a novel framework of attention-guided contrastive role representation learning for multi-agent reinforcement learning, ACORM. On the one hand, the role representation of each agent is inferred from the agent embedding through contrastive learning and clustering algorithms. On the other hand, the paper introduces an attention mechanism in value decomposition to enhance agent coordination in the role space. By introducing the above two contributions, ACORM performs better than other role-based multi-agent reinforcement learning algorithms in the SMAC environment. The paper also has intuitive visualizations to illustrate the role of the corresponding modules.

**Strengths:**

1. The paper is well-organized and easy to understand.
2. The experimental part has detailed case studies, which is very important for understanding the role of the two modules proposed in the paper. The figures about the t-sne embedding or weights corresponding to each snapshot have reasonable analysis.
3. The proposed framework is suitable for reinforcement learning algorithms based on value functions and those based on policy gradient. The relevant algorithms have been tested on SMAC and show that the ACORM variant is much better than the vanilla algorithm.
4. The proof of the ELBO is given in the appendix, which is correct to me and improves the soundness of the submission.

**Weaknesses:**

1. In cooperative multi-agent reinforcement learning, inferring the role of an agent based on its trajectory is not a novel method and has been proposed in many previous works [1, 2]. Moreover, none of the above-mentioned important papers are cited in the paper.
2. The number of baselines used for comparison with ACORM is relatively tiny. Why not use CDS [3] as a baseline, since you mentioned it in the paper?
3. SMAC is a relatively old multi-agent testbed. Recently, it has been pointed out that it has a series of problems [4]. I am not against the author evaluating the performance of the algorithm on SMAC, but I feel that the performance of the algorithm should be tested in multiple different domains. Many environments, such as the Google Research Football [5] mentioned in the paper, can be used to enhance the credibility of experimental results.
4. Ablation experiments are insufficient. Compared with the vanilla QMIX, ACORM_w/o_MHA still has an additional MLP and GRU for the global state. One wonders whether what really works is just the representation learning of the state trajectory before input to the Mixing Network.
5. It is not possible to reproduce the results from the description given in the paper. Some key details (such like $T_{cl}$) are unclear, and some key resources (code) are not furnished.



**Reference**

[1] Cao, Jiahang et al. LINDA: multi-agent local information decomposition for awareness of teammates. 2021.

[2] Yang, Jiachen et al. Hierarchical Cooperative Multi-Agent Reinforcement Learning with Skill Discovery. 2019.

[3] Li, Chenghao et al. Celebrating Diversity in Shared Multi-Agent Reinforcement Learning. 2021.

[4] Ellis, Benjamin et al. SMACv2: An Improved Benchmark for Cooperative Multi-Agent Reinforcement Learning. 2022.

[5] Kurach, Karol et al. Google Research Football: A Novel Reinforcement Learning Environment. 2019.

**Questions:**

Please see the questions in the weakness.

---

> ### Author Response · Authors · 2023-11-18
> **Response**
>
> **Q1: Experimental evaluation on more multi-agent testbeds.**
>
> A1: In the original version, we evaluated our method on six *representative* SMAC maps.
> Following your suggestion, we have conducted additional experiments on more SMAC maps and on Google research football environments, and provided the extended results in the updated Appendices B and C.
> The results on massively extended experiments are generally consistent with our previous conclusion that ACORM obtains the best performance in most scenarios and outperforms baselines by a larger margin on harder tasks.
>
>
> **Q2: About reproducibility and some experimental details (e.g., $T_{cl}$).**
>
> A2: Thank you for your careful reading.
> We have provided the source code of ACORM in the updated supplementary material to ensure reproducibility.
> In the original appendices, $T_{cl}$ is the time interval for updating contrastive loss as shown in Algorithm 1 and Tables 2 \& 4.
> To make these details clearer, we have modified these notations to be consistent across all appendices.
>
>
> **Q3: More sufficient ablation experiments, since ACORM\_w/o\_MHA still has an additional MLP and GRU for the global state.**
>
> A3: Thank you for your helpful advice.
> We have included an additional ablation baseline "ACORM\_w/o\_MHA (Vanilla)" that removes both the attention module and the additional GRU-MLP structure.
> In this way, the input to the mixing network is only the current global state $\bf{s}$$^t$, which is kept the same as the original QMIX algorithm.
> In Sec. 3.1 of the updated paper and the updated Appendix F, it can be observed that ACORM\_w/o\_MHA (Vanilla) obtains very similar performance compared to ACORM\_w/o\_MHA.
> It can be concluded that the effectiveness of our attention module comes from prompting the global state to attend to role representations other than the representation learning of the state trajectory via GRU.
>
>
> **Q4: The number of baselines is relatively tiny. Why not use CDS as a baseline, since you mentioned it in the paper?**
>
> A4: As mentioned in our original paper, there are a large number of related methods that involve agent heterogeneity and role emergence.
> For a moderate demonstration, we chose the basic QMIX and the four most relevant methods as our *five* baselines.
> We skipped some methods that generally perform worse than the selected baselines.
> For example, ROMA and RODE are both classical role-based methods, and RODE generally outperforms ROMA in the literature.
> Hence, we compare ACORM to RODE only.
> Further, we have included CDS as an additional baseline in the updated paper to make our evaluation more persuasive.
> Extended experimental results are consistent with our previous conclusion, and our method outperforms CDS in most scenarios.
>
>
> **Q5: More related works that infer the role of an agent based on its trajectory.**
>
> A5: We agree with you that inferring the role of an agent based on its trajectory has been investigated in many previous works, such as those role-based methods in our original paper.
> We have indeed included [2] as a Role Emergence method in the Related Work section (top, page 9) of the original paper, and we will also include [1] in our related work in the updated paper to make our literature review more complete and comprehensive.
> Compared to existing role-based methods, our method introduces two innovative components, i.e., the contrastive learning formulation for learning discriminative role representations and the attention mechanism to coordinate learned role representations for more expressive credit assignment, and our method obtains SOTA performance on most experiments compared to them.
>
> [1] Cao, Jiahang et al. LINDA: multi-agent local information decomposition for awareness of teammates, Science China Information Science 2021.
>
> [2] Yang, Jiachen et al. Hierarchical Cooperative Multi-Agent Reinforcement Learning with Skill Discovery, AAMAS 2020.

---

> ### Author Response · Authors · 2023-11-21
> **Your feedback is critical to us!**
>
> Dear reviewer,
>
> We were wondering if our response and revision have resolved your concerns. We are grateful for your recognition of the suitability and organization of our framework, the detailed case studies, and the soundness of the ELBO derivation.
>
> In our responses, we have **provided the source code** and **significantly extended experimental evaluation**, including experiments on more SMAC maps and **the Google research football testbed**, **an additional ablation study**, and **another baseline method**. We have **included more related work** on inferring an agent's role based on its trajectory.
>
> If our response has addressed your concerns, we would be grateful if you could re-evaluate our work.
>
> If you have any additional questions or suggestions, we would be happy to have further discussions.
>
> Best regards,
>
> The Authors

---

> > ### Comment · Reviewer_1yPm · 2023-11-21
> >
> > Thanks to authors for the detailed reply. I agree that the experimental results show the strong performance of ACORM. Many of my concerns were substantially addressed by authors, and I have updated my score. If more convincing results such as the performance of MAPPO-based ACORM are updated in the future, I will further improve my score.

---

> > > ### Author Response · Authors · 2023-11-22
> > > **Updated results on MAPPO-based ACORM in Appendix D.**
> > >
> > > Dear reviewer,
> > >
> > > Thank you for raising the rating of our work. We feel very encouraged by your reply. We are rushing to complete the evaluation results all these days. Following your suggestion, we have updated more experimental results on MAPPO-based ACORM in Appendix D (pages 19-20 in the updated main PDF), with a total of **six** representative maps: 2s3z, 3s5z, 5m\_vs\_6m, MMM2, corridor, and 3s5z\_vs\_3s6z. Akin to the QMIX-based version, ACORM improves the performance by the largest margin on super-hard maps that demand a significantly higher degree of behavior diversity and coordination. Extended experimental results consistently demonstrate ACORM’s superior learning efficiency in complex multi-agent domains.
> > >
> > > If our response has addressed your further concerns, we would be grateful if you could re-evaluate our work **again**. We will be extremely happy if you still have any additional concerns to share with us. Your opinion is critical to us. We will try our best to address your concerns.
> > >
> > > Best regards,
> > >
> > > The Authors

---

> ### Comment · Area_Chair_A6wJ · 2023-12-04
>
> Dear Reviewer 1yPm,
>
> As we progress through the review process for ICLR 2024, I would like to remind you of the importance of the rebuttal phase. The authors have submitted their rebuttals, and it is now imperative for you to engage in this critical aspect of the review process.
>
> Please ensure that you read the authors' responses carefully and provide a thoughtful and constructive follow-up. Your feedback is not only essential for the decision-making process but also invaluable for the authors.
>
> Thank you,
>
> ICLR 2024 Area Chair

---

### Official Review · Reviewer_WWCs · 2023-11-01

**Soundness:** 3 good
**Presentation:** 3 good
**Contribution:** 2 fair
**Rating:** 3
**Confidence:** 4

**Summary:**

This paper introduces the ACORM framework, which utilizes mutual information maximization to formalize role representation learning through a contrastive learning objective. It also incorporates an attention mechanism to encourage the global state to attend to learned role representations.
Empirical evaluations carried out on SMAC scenarios demonstrated that ACORM surpasses the performance of baseline methods. Additionally, visualizations and ablation studies show the pivotal roles played by the contrastive role representation and attention mechanism in this task.

**Strengths:**

* The proposed ACORM framework integrates representation learning, encoding the trajectory history from the traditional framework into a latent variable z. This representation associated with the role is learned through clustering and using positive-negative samples.
* Compared to the traditional framework, the global state used incorporates role-related representations through an attention mechanism.

**Weaknesses:**

* **Novelty and Reliability**: ACORM is not the first work to introduce the attention mechanism in the mixing network. For instance, works like [Qatten](https://arxiv.org/pdf/2002.03939.pdf) have introduced certain constraints in the network to satisfy the IGM principle. However, this paper does not provide evidence of complying with the IGM principle or any explanations.
* **Experimental Evaluation**: The experiments in the article are conducted solely on SMAC. To my knowledge, various versions of SMAC exist, and different algorithm implementations often involve custom modifications to this environment. Relying solely on SMAC for experiments may not be sufficiently persuasive. It might be beneficial to include experiments from other environments such as GRF and Ma-MuJoCo.
* **Reproducibility**: The supplementary materials do not include the source code, making reproducibility uncertain.

**Questions:**

* As previously mentioned, ACORM does not impose constraints on the attention mechanism, and it even utilizes the learned latent variable $z$. How can we ensure its correct execution under the CTDE paradigm?
* Regarding the analysis of Contrastive role representation, Figure 4 is not particularly convincing:
    * In subfigure (b), *(0) is even further from other points in its cluster compared to *(5). While I understand that clustering is done in higher dimensions, this example can be confusing.
    * While it's claimed that the role representation better forms coordination teams, in actuality, in subfigure (b) and (c), it seems just the agent embedding alone might suffice.
* Additional experiments are needed to bolster the paper's persuasiveness.
* The supplementary materials do not include the source code, making reproducibility uncertain.

I would like to raise my score if my concerns are addressed.

---

> ### Author Response · Authors · 2023-11-18
> **Response (Part 1/2)**
>
> **Q1: Reproducibility and more experimental evaluation.**
>
> A1: We have provided our source code in the updated supplementary material to ensure reproducibility.
> In the original version, we evaluated our method on six *representative* SMAC maps.
> Following your advice, we have conducted additional experiments on more SMAC maps and on Google research football environments, and provided the extended results in the updated Appendices B and C.
> The results of massively extended experiments are generally consistent with our previous conclusion that ACORM obtains the best performance in most scenarios and outperforms baselines by a larger margin on harder tasks.
>
> **Q2: More related work of introducing the attention mechanism in the mixing network.**
>
> A2: We agree with you that ACORM is not the first work to introduce the attention mechanism in the mixing network.
> As stated in Related Work of the original paper (bottom, page 9), attention has been widely applied to MARL domains for various utilities, including Qatten.
> However, our method differs largely from Qatten in how to use attention in the mixing network.
> Qatten proposes a novel value decomposition formulation that transforms individual Q-values to $Q_{tot}$ with a multi-head attention mechanism.
> That is, Qatten introduces a new mixing network structure and uses the attention mechanism *within* the mixing network.
> In contrast, we introduce the attention mechanism to incorporate role representations into the mixing network's input for better value decomposition.
> That is, we maintain the same mixing network architecture as QMIX and use the attention mechanism *at the input* of the mixing network.
> Further, our method mainly contains two innovative modules, i.e., the contrastive learning formulation for learning discriminative role representations and the attention mechanism to coordinate learned role representations for more expressive credit assignment.
> The two components work as a whole to efficiently incorporate the role information into MARL.
>
>
> **Q3: The compatibility with the IGM principle of ACORM.**
>
> A3: As claimed in A2, ACORM uses the same mixing network architecture as QMIX.
> Hence, ACORM's compatibility with the IGM principle is the same as that of QMIX.
> Formally, with QMIX's mixing network architecture, ACORM can enforce the monotonicity through the constraint on the relationship between $Q_{tot}$ and each $Q_a$ as $\frac{\partial Q_{tot}}{\partial Q_a}\ge 0, \forall a\in A$, and can naturally satisfy the IGM principle.
> Modifying the input to the mixing network (other than the mixing network structure itself) will not influence the monotonicity and the compatibility with the IGM principle.

---

> ### Author Response · Authors · 2023-11-18
> **Response (Part 2/2)**
>
> **Q4: ACORM's correct execution under the CTDE paradigm.**
>
> A4: As stated in A2 and A3, we use the same mixing network architecture as QMIX and just modify the input to the mixing network.
> During centralized training, we can access the global state $\bf{s}$ and role representations $z_i$ of each agent $i$.
> After prompting the global state to attend to role representations, we concatenate the attention output and the original global state as the input to the mixing network for value decomposition (where the hypernetworks produce the weights and biases for mixing network layers).
> During execution, each agent can make decisions according to the individual Q-network only, without access to the global state or others' role representations, i.e., decentralized execution.
> As we have emphasized in our original paper, ACORM strictly follows the CTDE paradigm and is fully compatible with CTDE methods.
>
>
> **Q5: In Fig. 4(b),  agent 0 is even further from other points in its cluster compared to agent 5.**
>
> A5: We cluster agent embeddings in an *unsupervised* way, and there is no ground truth to indicate the oracle cluster assignment.
> The effectiveness of our method relies on the mechanism of whether agents with similar behavior patterns tend to be grouped into the same cluster, and whether the contrastive learning objective can push points within the same cluster closer to each other and promote different clusters to be notably separated.
> Fortunately, our method meets this condition in most cases, as shown in Figs. 4-5.
> Several outliers, e.g., Agent 0 is even further from other points in its cluster compared to Agent 5 in Fig. 4(b), will not have a fundamental impact on the effectiveness of our method.
>
>
> **Q6. In Figs. 4(b) and 4(c), it seems that just the agent embedding alone might form good coordination teams.**
>
> A6: Thank you for your careful reading.
> Your observation totally matches the experimental analysis in our original paper (middle, page 7) as: "Initially ($t=1,12$), all agent embeddings tend to be crowded together with limited discrimination, and the K-means algorithm moderately separates them into several clusters. Via contrastive learning, the acquired role representations within the same cluster are pushed closer to each other, and those in different clusters are notably separated.
> At a later stage ($t=40$), agent embeddings are scattered widely throughout the space with a good clustering effect. This phenomenon indicates that the system has learned effective role assignment with heterogeneous behavior patterns.
> Then, the role encoder transforms these agent embeddings into more discriminative role representations."
> The discriminative patterns of agent embeddings in later stages could benefit from the early training of contrastive role representations.
> While the agent embedding alone might form good coordination teams at later stages, our contrastive learning module can help yield better coordination teams with more discriminative patterns.
> Ablation studies in Sec. 3.1 and Appendix F have also verified this point.

---

> ### Author Response · Authors · 2023-11-21
> **Your feedback is critical to us!**
>
> Dear reviewer,
>
> We were wondering if our response and revision have resolved your concerns. In our responses, we have **provided the source code** and **significantly extended experimental evaluation**, including experiments on more SMAC maps and the Google research football testbed, an additional ablation study, and another baseline method. We have **clarified ACORM's full compatibility with the IGM principle and the CTDE paradigm**.
>
> We are really looking forward to discussing this with you so that we could continually improve our work. Your feedback is critical to us.
>
> If our response has addressed your concerns, we would be very grateful if you could re-evaluate our work.
>
> Best regards,
>
> The Authors

---

> ### Author Response · Authors · 2023-11-22
> **Please share with us if you have any additional concerns!**
>
> Dear Reviewer,
>
> Apologies for reaching out once more, as today might be the final day of the discussion phase.
>
> We were wondering what could we do to further address your concerns for a higher rating. Our first guess on the reason that prevents you from a higher rating is that you might share the concerns on the reproducibility and experimental evaluation (also raised by two Reviewers 53j5 and 1yPm). We would like to raise your attention that Reviewer 1yPm replied to us very recently, approved **the significantly extended experimental results**, and increased the rating. We thus hope our responses to them could also solve your potential concerns.
>
> At the same time, we will be extremely happy if you still have any additional concerns to share with us. Your opinion is critical to us. We will try our best to address your concerns.
>
> Best regards,
>
> The Authors

---

> ### Comment · Area_Chair_A6wJ · 2023-12-04
> **[Important] Response Required to Authors' Rebuttal**
>
> Dear Reviewer WWCs,
>
> As we progress through the review process for ICLR 2024, I would like to remind you of the importance of the rebuttal phase. The authors have submitted their rebuttals, and it is now imperative for you to engage in this critical aspect of the review process.
>
> Please ensure that you read the authors' responses carefully and provide a thoughtful and constructive follow-up. Your feedback is not only essential for the decision-making process but also invaluable for the authors.
>
> Thank you,
>
> ICLR 2024 Area Chair

---

### Official Review · Reviewer_53j5 · 2023-11-08

**Soundness:** 3 good
**Presentation:** 2 fair
**Contribution:** 3 good
**Rating:** 5
**Confidence:** 3

**Summary:**

The paper introduces a method that combines a multi-head attention structure and the InfoNCE contrastive learning framework to enhance learning efficiency in MARL tasks by learning and utilizing role representations.

**Strengths:**

1.	The description of the methodology is clear and accurate.
2.	The performance of the experiments' results is really promising and impressive.
3.	The chapter of related works is rich and comprehensive.

**Weaknesses:**

1.	Extremely Lack of Experiments. There is a lack of experiments as the author only conducted experiments on 6 maps in SMAC. The same applies to the ablation experiments. Hope there will be additional experiments in a wider range of environments and on more maps within SMAC.
2.	The author did not provide the source code to verify.
3.	There might be some errors in the analysis. Such as the analysis in Appendix D, there is room for debate regarding the phenomenon of sub-groups. It is incorrect to measure the distance between 2 points within 1 cluster in the original space based on their proximity in the t-SNE space. Evaluating the emergence of sub-groups should start from the original space rather than the two-dimensional space after t-SNE reduction. Similarly, after t-SNE reduction, the distance between clusters still does not reflect the real distance. Therefore, the conclusion of 'their role representations are still closer to each other' in the later part still requires the author's reconsideration.
4.	The relationship between the two parts, MHA and CL, in the article is not particularly close; they seem more like two relatively independent components.
5.	There are some typos and the writing of the paper needs some improvement.

**Questions:**

1.	In Fig 4. (c), both Agent 5 and 8 are 'Dead Marines'. Why are they clustered into different classes?
2.	Why was a new map, 2s3z, introduced for the experiments in MAPPO? Why not directly use the previously employed 3s5z_vs_3s6z map? I would like to see an additional MAPPO experiment on 3s5z_vs_3s6z.
3.	How is the setting cluster_num = 3 applied on the map 2c_vs_64zg when there are only 2 agents available for control on this map? Besides, why didn't the performance decline since it forces the strategies of each agent to diverge as same as the experiment about cluster_num = 5 applied on the map 5m_vs_6m?
4.	In the derivation, both the approximation and logK indicate that a larger value of K yields better results. The experiments conducted in the selected maps have a limited number of agents. It is suggested to have more agents and experiment with larger values of K. For example, experiments with larger K values, such as K=2, 4, 8, 16, can be run in scenarios like 30m and bane_vs_bane.
5.	The paper doesn't explicitly clarify the difference between clustering directly on agent embeddings and on role representations. As it considers role representations to be more discriminative, it's important to further elucidate the necessity of obtaining discriminative representations through contrastive learning.
6.	The model has been added with a global state GRU and a MHA structure. Therefore it increase the number of parameters of the networks. It is recommended to conduct ablation studies with the same network size.

Minor: Bigger size of networks and additional contrastive learning procedure may limit the application. The theoretical derivation of Theorem 1. is very similar to previous work.

---

> ### Author Response · Authors · 2023-11-18
> **Response (Part 1/2)**
>
> **Q1: Lack of source code and experiments.**
>
> A1: We have provided the source code of our paper in the updated supplementary material to ensure reproducibility.
> In the original version, we evaluated our method on six *representative* SMAC maps.
> Following your suggestion, we have conducted additional experiments on more maps within SMAC and on Google research football environments, and provided the extended results in the updated Appendices B and C.
> The results of massively extended experiments are generally consistent with our previous conclusion that ACORM obtains the best performance in most scenarios and outperforms baselines by a larger margin on harder tasks.
>
> **Q2: It is suggested to have more agents and experiments with larger values of K.**
>
> A2: Following your suggestion, we have conducted additional experiments with larger numbers of agents, including the maps of 30m and bane\_vs\_bane.
> Results can be found in the updated Appendix B.
>
>
> **Q3: It is recommended to conduct ablation studies with the same network size, as our model is added with a global state GRU and an MHA structure.**
>
> A3: Following your advice, we have kept the network size of ACORM the same as all ablations for a fair comparison.
> Further, we have added another ablation baseline to identify the respective effects of the attention module and the state trajectory encoding via GRU.
> The updated experimental results are shown in Sec. 3.1 of the revised paper and in the revised Appendix F.
> It can be observed that the ablation performance is generally consistent with that of our original paper.
>
>
> **Q4: Analysis on the visualization of agent embeddings and role representations via t-SNE.**
>
> A4: We agree with you that after t-SNE reduction, the distance between clusters does not match the real *absolute* distance.
> However, what we care about is the *relative* distance between points or clusters.
> The t-SNE is a widely used dimensionality reduction technique for embedding high-dimensional data for visualization in a low-dimensional space, in such a way that similar objects are modeled by nearby points and dissimilar objects are modeled by distant points with high probability.
> Hence, the distance between points or clusters via t-SNE reduction can reflect their relative distances in the original space.
>
>
> **Q5: Why dead Marines 5 and 8 are clustered into different classes in Fig. 4(c).**
>
> A5: We cluster agent embeddings in an *unsupervised* way, and there is no ground truth to indicate the Oracle cluster assignment.
> The effectiveness of our method relies on the mechanism that agents with similar behavior patterns tend to be grouped into the same cluster.
> Fortunately, our method meets this condition in most cases, as shown in Figs. 4-5.
> Several outliers, e.g., dead Marines 5 and 8 are clustered into different groups, will not have a fundamental impact on the effectiveness of our method.
> As we can see in Fig. 4(c), Marines \{2,3,4,6,7\} are in the offense team, Marauders \{0,1\} and Marine 5 form one dead group, Marine 8 alone represents another dead group, and Medivac 9 alone forms the rescue group.

---

> ### Author Response · Authors · 2023-11-18
> **Response (Part 2/2)**
>
> **Q6: How is the setting cluster\_num=3 applied on the map 2c\_vs\_64zg? Besides, why didn't the performance decline since it forces the strategies of each agent to diverge?**
>
> A6: The number of clusters is less or equal to the number of agents.
> We have specifically pointed out that the cluster number is 2 on the 2c\_vs\_64zg map in the updated Appendices B and D.
> When each agent represents a distinct role cluster that forces the strategies of each agent to diverge, the performance may decline compared to those with effective team composition (e.g., K=5 compared to K=2,3,4 on the map 5m\_vs\_6m).
> However, it could still obtain improved performance compared to most baselines in Fig. 2, since it can explore cooperatively heterogeneous behaviors via our discriminative role representation learning.
>
>
> **Q7: Relationship between MHA and CL.**
>
> A7: We use CL to promote discriminative role representation learning, and utilize an MHA mechanism to coordinate learned role representations for more expressive credit assignment.
> The two innovative components work as a whole to efficiently incorporate the role information into MARL to promote behavior heterogeneity, knowledge transfer, and skillful coordination across agents.
>
>
> **Q8: Clarify the difference between clustering directly on agent embeddings and on role representations, and further elucidate the necessity of obtaining discriminative representations through contrastive learning.**
>
> A8: As we clarified in the 'Negative Pairs Generation' part of the original paper (bottom, page 4), we perform clustering according to agent embeddings, not role representations.
> First, we obtain agent embedding by encoding the agent's trajectory as $e_i^t=f_{\phi}(o_i^t, a_i^{t-1}, e_i^{t-1})$, and the agent embedding is transformed into role representation by a role encoder as $z_i^t=f_{\theta}(z_i^t|e_i^t)$ where the role encoder $\theta$ is optimized by a contrastive learning objective.
> As shown in the original paper, we have elucidated the necessity of obtaining discriminative representations through contrastive learning in Sec. 2.1.
> Further, experimental performance and ablation studies in Sec. 3.1 have also verified the effectiveness of our constrastive learning module, and visualization in Sec. 3.2 has clearly shown that the role encoder is capable of transforming agent embeddings into more discriminative representations.
>
>
> **Q9: There are some typos and the writing of the paper needs some improvement.**
>
> A9: Thank you for your careful advice. We have fixed several typos and improved the writing as much as we can.
>
> **Q10: Additional MAPPO experiment on 3s5z\_vs\_3s6z.**
>
> A10: In the original version, we evaluated MAPPO-based ACORM on three *representative* SMAC maps.
> Due to the very limited time for rebuttal revision, the performance of MAPPO-based ACORM on more SMAC maps is not updated.
> Debugging MAPPO-style algorithms is generally more time-consuming, since it involves many hyperparameters and empirical tricks.
> We hope that the extended experimental evaluation could demonstrate adequate persuasiveness of our method, and we are rushing to complete the evaluation results and trying to update them before the rebuttal deadline.

---

> ### Author Response · Authors · 2023-11-21
> **Looking forward to further discussions!**
>
> Dear reviewer,
>
> We were wondering if our response and revision have resolved your concerns. In our responses, we focus on ensuring the reproducibility of our work and significantly extending empirical evaluation, including experiments on more SMAC maps and the Google research football testbed, an additional ablation study, and another baseline method.
>
> If our response has addressed your concerns, we would be grateful if you could re-evaluate our work.
>
> If you have any additional questions or suggestions, we would be happy to have further discussions.
>
>
> Best regards,
>
> The Authors

---

> ### Author Response · Authors · 2023-11-22
> **Please share with us if you have any additional concerns!**
>
> Dear Reviewer,
>
> Apologies for reaching out once more, as today might be the final day of the discussion phase.
>
> We were wondering what could we do to further address your concerns for a higher rating. Our first guess on the reason that prevents you from a higher rating is that you might share the concerns on the reproducibility and experimental evaluation (also raised by two Reviewers WWCs and 1yPm). We would like to raise your attention that Reviewer 1yPm replied to us very recently, approved the **significantly extended experimental results**, and increased the rating. We thus hope our responses to them could also solve your potential concerns.
>
> At the same time, we will be extremely happy if you still have any additional concerns to share with us. Your opinion is critical to us. We will try our best to address your concerns.
>
> Best regards,
>
> The Authors

---

> ### Author Response · Authors · 2023-11-22
> **Updated results on MAPPO-based ACORM in Appendix D.**
>
> Dear reviewer,
>
> We are rushing to complete the evaluation results all these days. Promisingly, we have updated more experimental results on MAPPO-based ACORM in Appendix D (pages 19-20 in the updated main PDF), with **a total of six representative maps: 2s3z, 3s5z, 5m_vs_6m, MMM2, corridor, and 3s5z_vs_3s6z**. Akin to the QMIX-based version, ACORM improves the performance by the largest margin on super-hard maps that demand a significantly higher degree of behavior diversity and coordination. Extended experimental results consistently demonstrate ACORM’s superior learning efficiency in complex multi-agent domains.
>
> If our response has addressed your further concerns, we would be grateful if you could re-evaluate our work again. We will be extremely happy if you still have any additional concerns to share with us. We will try our best to address your concerns.
>
> Best regards,
>
> The Authors

---

> ### Comment · Area_Chair_A6wJ · 2023-12-04
> **[Important] Response Required to Authors' Rebuttal**
>
> Dear Reviewer 53j5,
>
> As we progress through the review process for ICLR 2024, I would like to remind you of the importance of the rebuttal phase. The authors have submitted their rebuttals, and it is now imperative for you to engage in this critical aspect of the review process.
>
> Please ensure that you read the authors' responses carefully and provide a thoughtful and constructive follow-up. Your feedback is not only essential for the decision-making process but also invaluable for the authors.
>
> Thank you,
>
> ICLR 2024 Area Chair

---

### Author Response · Authors · 2023-11-18
**Revision Summary**

We want to thank all the reviewers for their detailed comments and careful reading.
We summarize all the reviewers' concerns into four categories:
1) On the reproducibility and more experimental evaluations;
2) On more experimental baselines and ablation studies;
3) On ACORM's compatibility with the IGM principle and CTDE paradigm;
4) On more related work of introducing attention in the mixing network and inferring an agent's role based on its trajectory.

We have made a number of changes to address reviewers' suggestions and concerns.
Main modifications are marked with a blue color in the updated main paper and appendices.
A short summary of the modifications is made as

1. We **provide the source code** of our paper in the updated supplementary material to ensure reproducibility.
2. We significantly expand experimental evaluations to further demonstrate ACORM's SOTA performance, including **12 SMAC maps** (Appendix B), **three offensive scenarios on Goole research football** (Appendix C).
3. We conduct ablation studies with the same network size, and **add another ablation baseline** to identify the respective effects of the attention module and the state trajectory encoding.
4. We **include CDS [1] as an additional baseline** for experimental evaluation.
5. We include more references (e.g., [2]) to make the related work more complete and comprehensive.
6. We clearly clarify experimental details and modify some notations to be consistent across all appendices.
7. We fix several typos and improve the writing of the paper.

[1] Li, Chenghao et al. Celebrating Diversity in Shared Multi-Agent Reinforcement Learning. NeurIPS, 2021.

[2] Cao, Jiahan et al. LINDA: multi-agent local information decomposition for awareness of teammates. Science China Information Sciences, 2023.


In summary, most of the reviewers' comments were focused on experimental evaluation on more SMAC maps and other testbeds.
We would be grateful if the reviewers were affirmative to **ACORM on the methodology side and its impressive performance** (especially on super hard tasks), given that **ACORM: i) introduces two innovative components of contrastive role representation learning and attention-guided role coordination; ii) precisely formulates role representation learning with a sound ELBO derivation; iii) obtains state-of-the-art performance on many cooperative MARL tasks; and iv) exhibits detailed experimental case studies and reasonable analysis**.
We have significantly extended our experimental evaluation on more SMAC maps and on challenging Google research football environments, and we have included more sufficient ablation studies and another baseline method.
Impressively, the results of massively extended experiments are generally consistent with observations and conclusions from our previous paper.
Due to the very limited time for rebuttal revision: i) The performance of the new baseline CDS on additional 6 SMAC maps is not updated; ii) The performance of MAPPO-based ACORM on more SMAC maps is not updated; iii) In GRF environments, we only compare ACORM to QMIX and CDS, as other baselines (RODE, EOI, MACC, CIA) are not evaluated on GRF in their original papers.
We hope that the extended experimental evaluation could demonstrate adequate persuasiveness of our method, and we are rushing to complete the evaluation results and trying to update them before the rebuttal deadline.

Please let us know if we have addressed your concerns. We are more than delighted to have further discussions and improve our manuscript.
If our response has addressed your concerns, we would be grateful if you could re-evaluate our work.

Best regards,

The Authors

---

> ### Author Response · Authors · 2023-11-21
> **Looking forward to further discussions!**
>
> Dear reviewers,
>
> We were wondering if our responses and revision have resolved your concerns. In our responses, we have significantly extended experiment results and provided the source code based on your suggestions. Please let us know if we have addressed your concerns. We are more than delighted to have further discussions and improve our manuscript.
>
> Best regards,
>
> The Authors

---

### Meta-Review · Area_Chair_A6wJ · 2023-12-06

**Metareview:**

The paper under consideration introduces an innovative approach in cooperative multi-agent reinforcement learning, and proposes a framework with contrastive role representation learning and attention-guided role coordination. The paper sits at the boundary of acceptance. The authors have responded comprehensively and in detail to the reviewers' comments, and have expanded their experimental evaluation as well as providing the source code for verification. This proactive response and the additional experiments strengthen the paper's standing. Key Considerations:

**Justification For Why Not Higher Score:**

The paper is currently positioned at the boundary of acceptance, with an average score of 5.5 from the remaining two reviewers, excluding the scores from Reviewer WWC, who did not provide feedback, and Reviewer jtxX, whose review was deemed overly brief and uninformative. Despite this borderline situation, the thorough and responsive actions taken by the authors in addressing the reviewers' concerns and their significant effort in expanding the experimental validation and providing source code merit consideration for acceptance.

**Justification For Why Not Lower Score:**

Given the authors' detailed and comprehensive response to reviewer feedback, along with the expanded experiments and provided source code, the paper merits consideration for acceptance. It stands at the boundary due to its initial score, but the efforts made by the authors to address concerns and strengthen their work are commendable. The paper's approach and improved experimental validation contribute positively to its field.

---

### Decision · Program_Chairs · 2024-01-16

Accept (poster)